# ASTROCOMPRESS:
# A BENCHMARK DATASET FOR MULTI-PURPOSE COMPRESSION OF ASTRONOMICAL IMAGERY

**Tuan Truong**[*]  **Rithwik Sudharsan**[*]  **Yibo Yang**  **Peter Xiangyuan Ma**  **Ruihan Yang**
UC Irvine       UC Berkeley        UC Irvine     UC Berkeley          UC Irvine

**Stephan Mandt**                    **Joshua S. Bloom**
UC Irvine                            UC Berkeley; LBNL

## ABSTRACT

The site conditions that make astronomical observatories in space and on the ground so desirable—cold and dark—demand a physical remoteness that leads to limited data transmission capabilities. Such transmission limitations directly bottleneck the amount of data acquired and in an era of costly modern observatories, any improvements in lossless data compression has the potential scale to billions of dollars worth of additional science that can be accomplished on the same instrument. Traditional lossless methods for compressing astrophysical data are manually designed. Neural data compression, on the other hand, holds the promise of learning compression algorithms end-to-end from data and outperforming classical techniques by leveraging the unique spatial, temporal, and wavelength structures of astronomical images. This paper introduces AstroCompress (https://huggingface.co/AstroCompress): a neural compression challenge for astrophysics data, featuring four new datasets (and one legacy dataset) with 16-bit unsigned integer imaging data in various modes: space-based, ground-based, multi-wavelength, and time-series imaging. We provide code to easily access the data and benchmark seven lossless compression methods (three neural and four non-neural, including all practical state-of-the-art algorithms). Our results on lossless compression indicate that lossless neural compression techniques can enhance data collection at observatories, and provide guidance on the adoption of neural compression in scientific applications.

## 1 INTRODUCTION

Machine learning is having an increasingly large impact on natural sciences (Carleo et al., 2019; Wang et al., 2023). One of the primary hurdles of data-driven scientific discovery is bandwidth bottlenecks in data collection and transmission, particularly if the data is collected autonomously in high-throughput scientific instruments like telescopes or biological sequencing devices. Pairing this with the precision demands of many scientific domains, there is a significant interest for improved data compression methods across scientific domains.

A compression algorithm, or *codec*, is comprised of a pair of algorithms that can encode data into a smaller size, and then decode back to the original or near-original data. Neural compression algorithms (Yang et al., 2023b) have recently surpassed traditional codecs on image (Ballé et al., 2017; Yang et al., 2020b; He et al., 2022) and video compression (Lu et al., 2019; Agustsson et al., 2020; Yang et al., 2020a; Mentzer et al., 2022). Compared to these visual media applications, where fast decoding is crucial, (Minnen, 2021), scientific domains often prioritize compression performance and/or encoding speed. This is because data analysis can be done on powerful supercomputers over days or weeks, but the instruments' data collection rates are extremely high and data must be transmitted rapidly. Science can more readily and substantially benefit from neural compression—which generally achieves high compression rates at the expense of time (Yang & Mandt, 2023).

---

[*]Equal contribution first authors. Correspondence to tuannt2@uci.edu, rithwik@berkeley.edu.

Scientific data exhibit unique statistical patterns and signal-to-noise distributions, potentially making traditional handcrafted codecs less suitable. Neural compression, being learned end-to-end from the data, can automatically exploit redundancies occurring in the data and accelerate the development of practical codecs for custom applications.

**We are at the cusp of a scientific data explosion.** Current imaging efforts mapping the mouse hippocampus (Google, 2024) are estimated to contain 25 petabytes of imaging data. Human genomics data is expected to generate 2–10 exabytes of data over the next decade (NIH, 2024). The Square Kilometer Array (SKA), a ground-based telescope, is expected to collect 62 exabytes per year (Farnes et al., 2018), where improvements in lossless compression could significantly reduce storage costs.

In 2027, NASA will launch the Nancy Grace Roman Space Telescope, or "Roman." A recent audit (NASA, 2024) emphasizes that the single greatest concern for Roman is data transmission issues due to an "unprecedented" data scale and unprepared downlink networks that lack the necessary bandwidth. Space telescopes suffer a unique problem: due to limited bandwidth and onboard storage, any data that cannot be transmitted must be deleted and is thus permanently lost. Given Roman's estimated cost of \$4.3 billion, a mere $1\%$ improvement in lossless compression for Roman thus imputes a value of \$43 million on the additional data gathered. This will be the first space telescope with the ability to run modern codecs, as the recently launched James Webb Space Telescope (JWST) was planned more than 20 years ago and has highly limited compute onboard (Gardner et al., 2023).

Astrophysics presents novel challenges and opportunities for learned compression. Astronomical imaging captures the locations, spatial extent, temporal changes, and colors of celestial objects and events. Despite variations in detector and instrumentation physics across the electromagnetic spectrum, much of the raw data is stored as 2D digital arrays, mapping intensity at pixel locations $(x, y)$ to a projection of the flux of objects on sky. Repeated integrations enhance depth (higher signal-to-noise) through post-processing co-addition and can be used to measure time variations in celestial events. Color information is obtained from co-spatial observations using different instruments or filters with selective wavelength sensitivity.

As detector sizes grow and costs per pixel decline, larger optical and infrared arrays are being deployed. The Rubin Observatory, now in commissioning, has the largest optical camera ever built, with 189 CCDs comprising 3.2 billion pixels (Kahn et al., 2010). It can generate 20–30 TB of raw imaging data per night with 5-second integrations. The main imaging camera of JWST features ten $2048 \times 2048\,\mathrm{pix}^2$ detectors (Gardner et al., 2023), compared to the single $256 \times 256\,\mathrm{pix}^2$ array of the Hubble Space Telescope (Thompson et al., 1998). Unlike CCDs and CMOS detectors, which are read only once per exposure, IR arrays (e.g. that of JWST) are continuously sampled during integration, producing 3D data cubes with arbitrarily large temporal samples. Optical and IR instruments convert photon signals captured by pixelated semiconductor detectors into digital values using analog-to-digital (A/D) converters, resulting in 16-bit depth images. Each pixel value includes flux from astronomical sources (which we refer to as "sources" in the rest of the paper), sky background, A/D-introduced noise, and "dark current" arising from the non-zero detector temperature.

The need to transmit large amounts of data efficiently and losslessly is a critical challenge for premier astronomical facilities, which often operate in remote locations to optimize observations. This remoteness complicates data transmission to distant computational and archival centers. For ground-based facilities, the inability to transmit raw data as quickly as it is obtained (e.g., via wired internet) means hard copies of the data must be (periodically) moved physically, degrading time-sensitive science (Bezerra et al., 2017). Space-based facilities risk losing data if it cannot be transmitted with high-enough bandwidth. These constraints are practical and evident in current space missions. The Kepler mission (Borucki et al., 2010) produced ~190 MB of raw data from 42 CCDs every 6 seconds but had to coadd 10–250 exposures and transmit only selected pixels around 200,000 preselected stars due to bandwidth and storage limits (Jenkins & Dunnuck, 2011). The TESS satellite (Ricker et al., 2015) combines 2-second integrations from 4 CCDs into frames of 60 seconds or 30 minutes and transmits only the $10 \times 10$ pixels around pre-designated objects and full-frame 30-minute coadds. JWST enforces strict data rate limits during observations to stay within deep space downlink constraints, affecting data collection and storage (STSCI, 2024). Bottlenecks in data transmission directly impact the scientific potential of modern observatories. Given the high cost of developing and maintaining modern observatory instruments, improved data compression can significantly increase the value of data available to researchers. Lossless compression of raw imaging

data before transmission is thus highly desirable, especially for space-based facilities, balancing improved compression ratios against computational and hardware costs.

This benchmark serves multiple purposes. Firstly, we release several large, unlabeled astronomy datasets spanning the range of imaging found across modern astronomy. We note that astronomy has several other modes of data collection, such as radio interferometry and spectroscopy, both of which are out of scope for our work. Our goal is to foster a machine-learning community focused on astrophysical data compression, aiming for improved compression methods that can eventually be practically deployed on space telescopes. Secondly, we benchmark several neural lossless compression algorithms on the data, demonstrating that neural compression has significant potential to outperform classical methods. We hope that our initial results will encourage further research (cf. Thiyagalingam et al. 2022), leading to even better outcomes and a deployable algorithm. Future astronomy codecs can also be developed and benchmarked on this dataset for *lossy* compression, where we anticipate significant progress and improvements.

In sum, our main contributions to the datasets and benchmarks track are as follows:

- A large ($\sim$320 GB), novel dataset captures a broad range of real astrophysical imaging data, carefully separated into train and test sets, with easy access via HuggingFace `datasets`.

- An extensive comparison of lossless classical and neural compression methods on this data, the first publication to our knowledge where neural compression has been systematically studied on astronomical imaging.

- Various qualitative analyses that further our understanding of the bit allocation in astronomical images and inform potential exploration for future *lossy* compression codec designs.

## 2 BACKGROUND AND RELATED WORK

Compression is widely used in astronomy to transmit raw data from satellites, to store smaller files in archives, and to speed the movement of files across networks. The consultative Committee for Space Data Systems (CCSDS, 2024) periodically reports on recommendations for methods—all variants of JPEG-LS and JPEG-2000 (see below), though not yet JPEG-XL—to compress images and data cubes as they are transmitted in packets from satellites to ground stations. Specialized commercially available space-hardened hardware modules that implement these standards are in use both by NASA and the European Space Agency (ESA). Once on the ground, images are usually held and transmitted by major archives with bzip2, Hcompress, or Rice compression (cf. Pence et al. 2009 and §4.1). Compressed image storage (and manipulation codes; Seaman et al. 2010) are standardized in the file formats, like FITS (Pence et al., 2010) and HDF5, commonly in use by astronomers. Many have studied and proposed refinements of these methods both for lossless (Villafranca et al., 2013; Pata & Schindler, 2015; Thomas et al., 2015; Maireles-González et al., 2023; Mandeel et al., 2021) and lossy (Maireles-González et al., 2022) compression. Notably, all of these works rely on manually designed codecs using relatively simple probability models and classical transforms from signal processing.

In contrast to traditional hand-designed codecs, neural compression algorithms based on deep generative models (Yang et al., 2023b) can be optimized end-to-end on the target data of interest, and has demonstrated significantly improved compression performance. Lossless compression involves estimating a probability model of the data, and neural lossless methods are typically built on discrete normalizing flows (Hoogeboom et al., 2019), diffusion models (Kingma et al., 2021), VAEs (Townsend et al., 2019; Mentzer et al., 2019), and probabilistic circuits (Liu et al., 2022). Unlike lossy compression which requires evaluating the (lossy) reconstruction quality, lossless compression is primarily concerned with the bit-rate (the lower the better) or compression ratio (the higher the better); however building a neural lossless codec with high compression ratio and low computation complexity remains a challenge.

A small but growing community has been testing neural compression methods for data from specialized scientific domains. Hayne et al. (2021) published a study on neural compression for image-like turbulence and climate data sets using a lossy neural compression model. Choi et al. (2021) studied similar neural compression models on plasma data. Huang & Hoefler (2023) compress climate data by overfitting a neural network and using the network weights as compressed data representation.

Wang et al. (2023) adopted a classical-neural hybrid approach in medical image compression. Overall, we are unaware of any efforts applying neural compression to astronomical images.

While the astronomical data in public archives (MAST, 2024) are vast, the assembly process for a machine learning suitable corpus requires significant domain knowledge. With dozens of parameters defining each observation (sky region, exposure time, filter, grating, pupil, etc.) and myriad data structures, these archives are too unwieldy for ML practitioners. Previous attempts at ML-friendly corpus creation, such as Galaxy10 (Leung & Bovy, 2019) (see also Xue et al. 2023 and Khujaev et al. 2023), have typically rescaled data to 8-bit RGB images, significantly reducing dynamic range and thereby losing much of the information critical for novel scientific analysis. Hayat et al. (2021) assembled a 266 GB corpus of processed `float32` $64 \times 64$ pix$^2$ 5-filter image cutouts around bright galaxies. AstroCompress, in contrast, is focused on `uint16` images that represent a much wider diversity of real-world raw data, including large regions of low SNR and major imaging artifacts.

## 3 ASTROCOMPRESS CORPUS

Our central contribution is the AstroCompress corpus, curated to capture a broad range of real astrophysical imaging data and presented to enable the exploration of neural compression. The corpus is released on HuggingFace and can be easily accessed using Python, with code examples in the Supplementary Material. The corpus consists of 5 distinct datasets, spanning a variety of observing conditions from space and from Earth, types of detector technology, and large dynamic ranges. The quantity of data is three orders of magnitude larger and more varied than previous compression-focused corpora (Pata & Schindler, 2015; Maireles-González et al., 2023) to ensure ample training data for ML-based approaches. In contrast to previous corpora containing only ground-based data, our dataset has a strong focus on space-based data, for which improved compression is much more critical. Besides the typical 2D imaging data, we also include higher-dimensional (3D and 4D) data cubes containing multiple images of the same spatial origin but along different wavelength and/or temporal dimensions. The raw data source for the data cubes provides only single timestep images, which were then scraped, mosaicked spatially to create larger images, and then stacked across time. These data cubes are a unique feature of our corpus, offering compression algorithms the opportunity to exploit redundancies along additional dimensions to achieve higher compression ratio, and enables the exploration of sequential compression techniques such as residual coding and adaptive coding (see Section 4). To help avoid over-fitting (or over-testing) on certain regions of the sky, great care was taken to ensure no two images in the same dataset overlap spatially. We briefly describe the five datasets comprising AstroCompress, presenting the some key features in Fig. 1, and defer details of their composition and acquisition to the Supplementary Material:

**GBI-16-2D (Keck)**    This is a diverse, 2D optical imaging dataset from the ground-based W. M. Keck Observatory. It contains 137 images of size either $2248 \times 2048$ or $3768 \times 2520$ pix$^2$, obtained in a variety of observing conditions, filters, and across exposure times from seconds to $>10$ min.

**SBI-16-2D (Hubble)**    This dataset is derived from the Hubble Space Telescope (HST) Advanced Camera for Surveys (ACS; Sirianni et al. 2005) observations in the F606W (~red) filter. It contains 4282 images of size $4144 \times 2068$ pix$^2$. A major challenge (and opportunity for compression) in these raw space-based images are the preponderance of random cosmic ray-affected pixels and charge transfer inefficiencies causing vertical stripping (see Fig. 1).

**SBI-16-3D (JWST)**    This dataset comes from the NIRCAM instrument onboard the James Webb Space Telescope (JWST), taken with the F200W (infrared) filter. It contains 1273 3D cubes of $T \times 2048 \times 2048$, for time steps $T$, typically ranging as $5 \lesssim T \lesssim 10$. The instrument repeatedly measures the cumulative optimal charge across time, and therefore the pixel value at a given spatial location cube increases with $T$ until reaching a saturation value ($2^{16} - 1$). This directly allows for residual coding, i.e., compressing an initial 2D frame and the temporal differences of 2D frames subsequently.

**GBI-16-4D (SDSS)**    This ground-based dataset is assembled from the Sloan Digital Sky Survey (SDSS; York et al. 2000). We assembled 500 four-dimensional cube representations of different $800 \times 800$ pix$^2$ portions of the sky, each one observed from $t = 1$ up to $T \approx 90$ times in $F$=5 filters $(u, g, r, i, z)$, each cube having shape $T \times F \times 800 \times 800$. Compared to JWST, this dataset contains

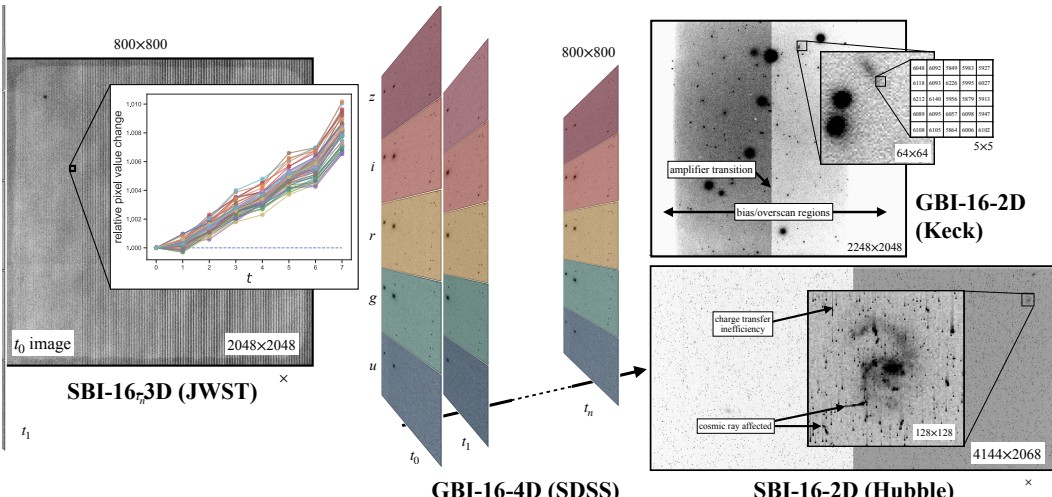

Figure 1: Depiction of salient features in the AstroCompress corpus using representative images from each dataset. Inset to the JWST $t_0$ (first) image are the value changes in time for a small sample of pixels. In SDSS there are 5 filtered images per observation epoch, up to a variable number $n$ observations in the same portion of the sky. The inset of Hubble zooms in on a spiral galaxy, showing cosmic ray hits (black) and charge transfer inefficiency, causing vertical flux smearing. The actual pixel values in Keck are shown for a zoomed-in $5 \times 5$ pix$^2$ region.

multiple channels associated with different wavelength filters, and presents rich possibilities for modeling and/or compression that takes into account correlation among all four dimensions.

**GBI-16-2D-Legacy** This small ground-based dataset obtained on multiple CCDs across many different telescopes is reproduced from the public corpus released by Maireles-González et al. (2023) and assembled by us in HuggingFace dataset format. Our experiments only made use of the subset of data from the Las Cumbres Observatory (**LCO**).

## 4 EXPERIMENTS

We establish the compression performance of our selected neural and non-neural compression methods on our proposed datasets. We describe our experiment protocol in Sec. 4.2, and present our main results in Sec. 4.3. Our results show that, even with only minimal architectural adjustments, neural compression can match or even surpass the best classical codecs. Our bit-rate estimates obtained with VDM (Kingma et al., 2021) consistently dominate all the neural and non-neural methods considered, suggesting significant room for future improvements. On the other hand, neural codecs designed for natural image data, such as L3C and PixelCNN, have difficulty exploiting cross-frame correlations in astronomy images, likely due to the image noise characteristics.

To better understand how the behavior of the algorithms depends on data characteristics, we examine bit-rate allocation both qualitatively and quantitatively in Sec. 4.4. Echoing earlier findings from Pence et al. (2009), we see a strong correlation between bit-rate and measures of noise such as SNR and exposure time, confirming that most of the bits are allocated to noisy pixels that can be well modeled by i.i.d. white noise distributions. Lastly, we explore the out-of-domain generalization performance of a neural compression method, Integer Discrete Flows, in Sec. 4.5.

### 4.1 COMPRESSION METHODS

We note that astronomy-specific compression implementations have stalled since Pence et al. (2009), and so the latest algorithm in current use is Rice. The most recent works by (Maireles-González et al., 2023) and the CCSDS (CCSDS, 2022) have established JPEG-2000 as a state-of-the-art, though it has yet to be deployed to telescopes. Members of our team regularly work with astronomy data collections pipelines, and we confirm this information.

We consider four non-neural methods as baselines, including three standard codecs from the Joint Photographic Experts Group (JPEG) and one codec developed by the Jet Propulsion Laboratory (JPL). The `imagecodecs` library provides the necessary APIs for all methods. Specifically, we run **JPEG-XL**, **JPEG-LS**, **JPEG-2000**, and **RICE** codecs in lossless mode with default settings. Additionally, we run JPEG-XL under standard and maximum effort modes as an extra reference, an algorithm that has not been tested for astronomy in any previous works, and show that it establishes a new state-of-the-art amongst non-neural methods.

We adopt three well-known *practical* neural lossless compression methods in the literature, representing key approaches in deep generative modeling for compression:

**Integer Discrete Flows (IDF) (Hoogeboom et al., 2019):** a flow-based model extending the concept of normalizing flows (Rezende & Mohamed, 2015) for lossless compression. Unlike conventional normalizing flow models that operate on continuous data, IDF employs discrete bijective mappings using invertible neural networks to connect discrete pixels with a discrete latent.

**L3C (Mentzer et al., 2019):** a VAE-based lossless compression method utilizing a two-part code (Yang et al., 2023b). It trains a hierarchical VAE with discrete latents; a given image is compressed by first entropy coding the inferred latents, and then entropy coding the image conditioned on the latents.

**PixelCNN++ (Salimans et al., 2017):** an autoregressive model using masked convolutions to model the distribution of each pixel given previous pixels in a raster scan order. PixelCNN++ naturally allows for lossless compression using autoregressive entropy coding (Mentzer et al., 2019).

Additionally, we use **Variational Diffusion Model (VDM) (Kingma et al., 2021)** to demonstrate the theoretical performance achievable by current state-of-the-art likelihood-based models. Unlike most diffusion models that target sample quality, VDM incorporates a learned noise schedule and Fourier features, and surpasses Transformer-based autoregressive models on likelihood scores. The likelihood score of VDM can be operationalized as the lossless compression cost of bits-back coding (Townsend et al., 2019; Kingma et al., 2021), although the resulting codec requires a high number of diffusion steps for encoding/decoding and may not be practical.

**Handling 16-bit data**: Most neural lossless compression methods are designed for RGB image compression, operating on 8-bit (unsigned) integers. To accommodate 16-bit data of AstroCompress, we minimally modify the neural compression methods as follows: for L3C and PixelCNN++, which both use a discretized logistic mixture likelihood model (Salimans et al., 2017), we increase the number of bins from $2^8 - 1$ to $2^{16} - 1$; for IDF, we simply change the input normalization constant from $2^8$ to $2^{16}$. Alternatively, we consider treating each 16-bit pixel as two sub-pixels: the most significant byte (MSB) and the least significant byte (LSB), converting a 1-channel 16-bit image into a 2-channel 8-bit image; we then double the number of input channels of each model accordingly. We refer to Supplementary Material for more details.

## 4.2 EXPERIMENT SETUP

We experiment on two categories of data: single-frame images and spectrally/temporally correlated images captured at multiple wavelengths or time steps. We consider the LCO, Keck, and Hubble datasets as single-frame image datasets. For the JWST datasets, we select the first time step of each 3D image cube and form a single-frame dataset, called JWST-2D, and the residual (i.e., difference) between consecutive frames as a separate, temporally correlated dataset, called JWST-2D-Res. We sub-select three benchmark datasets from SDSS as follows: (1) the first time step of the $r$ filter band forms a single-frame dataset, called SDSS-2D; (2) the first time step of the $g$, $r$, and $i$ filter bands constitute a spectrally correlated dataset, called SDSS-3D$\lambda$; and (3) the first three time steps of the $r$ filter band create a temporally correlated dataset, called SDSS-3DT.

For all experiments, we use a fixed split of training and testing images (details in Supplement). The training set is further divided into 85% for training and 15% for validation. For each method, we train and evaluate two model variants as described above, either handling 16-bit input directly or treating it as 8-bit input with double the number of channels; we report the best compression performance between the two variants. We train on random $32 \times 32$ spatial patches and apply random horizontal flipping. For evaluation, we divide each image evenly into $32 \times 32$ patches, apply reflective padding beforehand if needed. We evaluate the model's compression performance by compressing all patches

| Experiment | Neural Methods | | | | Non-neural Methods | | | | |
|---|---|---|---|---|---|---|---|---|---|
| | IDF | L3C | PixelCNN++ | VDM | JPEG-XL (max) | JPEG-XL | JPEG-LS | JPEG-2000 | RICE |
| LCO | 2.83 | 1.67 | 2.02 | **3.64** | 2.98 | 2.78 | 2.81 | 2.80 | 2.65 |
| Keck | 2.04 | 1.89 | 2.08 | **2.11** | 2.01 | 1.97 | 1.97 | 1.96 | 1.84 |
| Hubble | 2.94 | 2.90 | 3.13 | **3.33** | 3.26 | 2.92 | 2.86 | 2.67 | 2.64 |
| JWST-2D | **1.44** | 1.38 | **1.44** | **1.44** | 1.38 | 1.33 | 1.35 | 1.37 | 1.24 |
| SDSS-2D | 2.91 | 2.36 | 3.35 | 3.27 | **3.38** | 3.14 | 3.16 | 3.20 | 2.96 |
| JWST-2D-Res | 3.14 | 2.91 | 2.80 | — | **3.35** | 2.37 | 3.24 | 1.69 | 3.08 |
| SDSS-3Dλ | 3.05 | 2.29 | 2.88 | — | **3.49** | 3.23 | 3.24 | 3.28 | 3.05 |
| SDSS-3DT | 3.03 | 2.59 | 3.02 | — | **3.48** | 3.23 | 3.24 | 3.29 | 3.05 |

Table 1: Compression ratios for all methods across experiments, with bold text indicating the best performance and underlined text indicating the second best. The top and bottom subsections of the table contain single-frame and spectrally/temporally correlated-frame compression results, respectively.

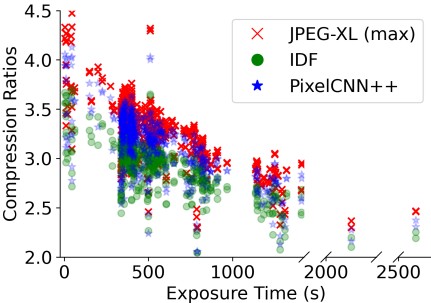

Figure 2: Hubble exposure times plotted against compression ratios using various algorithms. Longer exposure times tend to induce more incompressible noise and, hence, reduce compression ratios.

of an image and combining the results to determine overall performance on that image. This process is then repeated for all test images, and the average compression performance is reported. Compression ratio is calculated as the uncompressed bit depth / negative log-likelihood assigned by the model, aligning closely with actual on-disk entropy coding performance using arithmetic coding.

## 4.3 COMPRESSION PERFORMANCE

The top subsection of Table 1 presents compression ratios on single-frame compression experiments. Among the neural codecs, PixelCNN++ and IDF consistently achieve the most competitive performance. The estimated compression performance of VDM significantly surpass all existing methods on most datasets, suggesting significant room for future improvement with neural compression methods. Among the non-neural codecs, JPEG-XL (max) dominates across all datasets, establishing itself as the new state-of-the-art non-neural method which has not been considered in prior literature. Note that on the LCO dataset, the previous highest compression ratio was 2.79 by Maireles-González et al. (2023), and our results for JPEG-LS, JPEG-2000, and RICE, are consistent with those of Maireles-González et al. (2023) on this dataset.

When it comes to spectrally/temporally correlated data, we expect higher compression ratios for all the methods due to the additional correlations that can be exploited. This is indeed the case, as seen in the bottom subsection of Table 1; however, non-neural codecs show surprisingly superior performance boosts than the neural codecs. This indicates a need for further improvement in the neural compression techniques, which have largely been designed for image compression, to better extract cross-wavelength or cross-timestep information.

## 4.4 THE EFFECT OF NOISE

Following Pence et al. (2009), our Figure 2 illustrates an inverse relationship between compression ratios and exposure time, which is one of the key variables in determining the signal-to-noise ratio (SNR) in an image. We demonstrate a negative correlation between these two variables, likely because

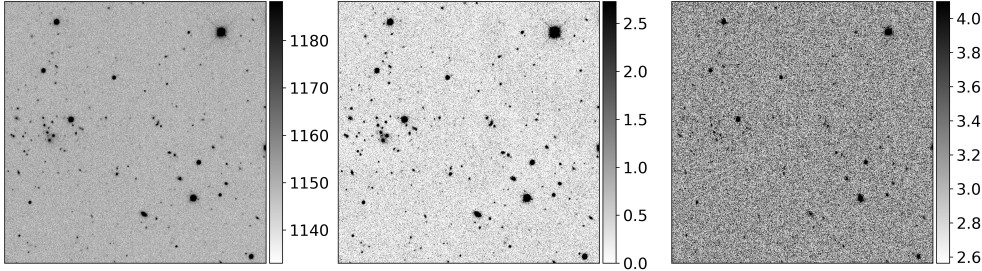

Figure 3: From left to right, an example SDSS-2D image: raw image, SNR heatmap, and PixelCNN++ bitrate heatmap. Colors are z-score normalized for visualization; colorbars indicate true values.

an increase in exposure time results in an increased number of noise bits. We hypothesize that the plateau seen will reverse at higher exposure times as many CCD pixels reach their max physical value, reducing image entropy. In Supplementary Section D, we further explore a strong relationship between background pixel noise levels and compressibility amongst all test images from all datasets.

Figures 3 and 4 use data from a single, representative SDSS-2D frame. We used `photutils` (photutils, 2024) estimate the sky's background noise to get the SNR at each pixel, and then to compute a mask of all sources. The 2D background was estimated by dividing the $800 \times 800$ image into $50 \times 50$ patches and excluding pixels above $3\sigma$ of the median value. The medians and standard deviations of the remaining pixels were interpolated to get the final background and noise images. SNR was calculated as `SNR = (original_image - background) / noise`. Source pixels were detected by applying a kernel around any pixels above $3\sigma$, creating a smoothed mask for signal-generating objects.

Figure 3 demonstrates that source pixels have higher bitrates, as expected—in a sense, these pixels are more "surprising," and thus a lower likelihood is assigned. Interestingly, the background pixel regions of the bitrate heatmap are significantly more noisy than the corresponding SNR background. This suggests that there may be potential for reduction in the bitrate of the higher bitrate background pixels, as we might expect most background pixels to exhaust a similar number of bits.

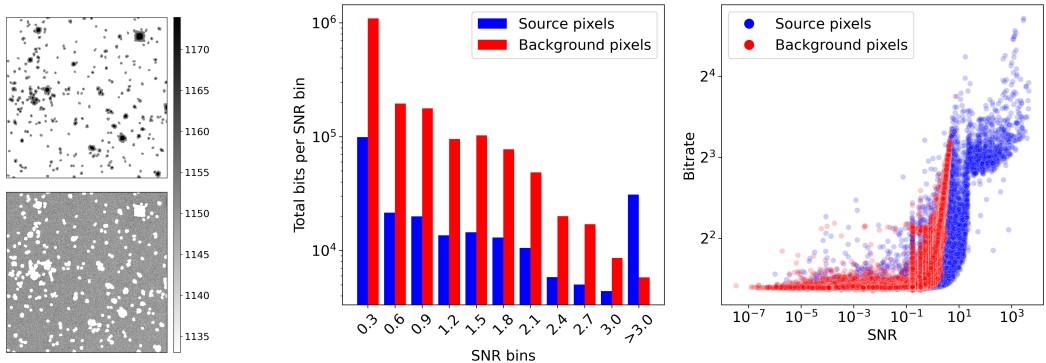

Figure 4: **Left**: SNR heatmap of an example image (Figure 3), showing source (top) vs. background (bottom) pixels. **Middle**: histogram of PixelCNN++ total bit allocation for various binned SNR values. **Right**: scatterplot showing the positive correlation between SNR and PixelCNN++ bitrate.

Figure 4 furthermore demonstrates the extreme bit-rate consumption by background noise pixels. On this image, 98.5% of pixels fell under 3 SNR. Some source pixels were placed in the lowest SNR bin, but this is likely due to some overestimation of source radii in the source masking process. Interestingly, the scatterplot resembles a step function, with a jump from `Bitrate` $\approx 3$ bits per pixel to `Bitrate` $\approx 10$ bits per pixel at `SNR` $\approx 10^1$—the transition point where stars and galaxies emerge over background noise.

## 4.5 GENERALIZATION PERFORMANCE

We take IDF as our representative model for investigating generalization performance, where we train and evaluate on all pairs of the single-frame compression tasks, as well as train on all these datasets combined. Table 2 shows that the cross-data generalization performance depends heavily on the training data used, and training on some of the datasets (e.g., Keck, JWST-2D) consistently resulted in better generalization performance than others (LCO, Hubble). In particular, the test performance on SDSS-2D when trained on Keck was even better than when trained on SDSS-2D itself. We hypothesize that the diversity of wavelength filters

|  | LCO | Keck | Hubble | JWST-2D | SDSS-2D |
|---|---|---|---|---|---|
| LCO | **2.83** | 1.01 | 1.09 | 0.84 | 2.31 |
| Keck | 2.70 | **2.05** | 2.20 | 1.19 | 3.02 |
| Hubble | 0.67 | 0.94 | 2.94 | 1.22 | 0.69 |
| JWST-2D | 1.46 | 1.45 | 1.47 | **1.44** | 1.50 |
| SDSS-2D | 2.27 | 1.24 | 1.75 | 1.02 | 2.91 |
| All data | 2.82 | 1.87 | **2.98** | 1.38 | **3.18** |

Table 2: IDF generalized performance across single-frame datasets. Rows indicate train set; columns indicate test set. Bold indicates best in the test set; underline indicates second-best.

and background noise in the Keck dataset, as mentioned in Section 3 and shown in Figure 5, may explain its unexpected generalizability. Overall, the best generalization performance was achieved by training on all the datasets combined, suggesting the importance of exposing the model to a variety of data characteristics. Additional model experiments trained on RGB images and evaluated on our dataset are discussed in the supplementary section E.

## 4.6 COMPUTATIONAL METRICS

To offer practical considerations, we present runtime metrics that would be valuable in assessing the feasibility of these methods for real-world applications. Table 3 shows various runtime for inference from neural methods and coding time for classical methods. JPEG-XL with max effort and JPEG-2000 seem to scale quadratically with the number of input pixels, while all other neural and non-neural algorithms scale linearly with the number of input pixels. Note that in order to work with limited GPU memory, our neural methods primarily operate on $32 \times 32$ patches, which very likely limits the achievable compression ratio. By comparison, our non-neural baselines always receive full images as input. We stress that the JPEG-XL max algorithm takes nearly 90 seconds for a Hubble image, which would likely be infeasible for practical use given that Hubble collects 15-30 GB of data per day, nearly requiring the entire day for compression alone (NASA, 2014). IDF is roughly 10x faster than JPEG-XL max and often outperforms all non-neural methods other than JPEG-XL max. While we acknowledge that runtimes are inherently hardware and parallelization-dependent, this work is primarily aimed at inspiring a broader research trajectory towards advanced compression techniques over the coming decade. Consequently, we defer the nuanced practical considerations of space-compatible hardware and runtime optimization to future investigations, noting that all presented algorithms leverage unoptimized reference implementations. We note that the long runtime of VDM is currently impractical due to a large number of expensive neural network evaluations. However, there exists extensive research on speeding up diffusion models by several orders of magnitude Salimans & Ho (2022) Cao et al. (2024) Ulhaq & Akhtar (2022) Yang et al. (2023a), which can potentially translate their excellent bit-rate estimates into practical compression performance.

## 5 DISCUSSION OF FUTURE DIRECTION

**Lossy compression in astronomy** can likely achieve significantly higher compression ratios compared to lossless methods. For astronomy, where most pixels are dominated by noise, lossy compression is particularly promising. Collaborations between astronomers and machine learning experts could lead to the development of advanced lossy compression algorithms that selectively discard non-essential data, preserving only the scientifically valuable information. One simple approach might mask astronomical sources and prioritize the accuracy of source pixels. Another approach could involve near-lossless compression, which ensures a strict user-defined upper bound on the error of every individual reconstructed pixel (Bai et al., 2022). Such an approach would be attractive to astronomers, who may desire a known error measurement for uncertainty propagation.

**Encode-decode time tradeoffs** factor crucially in remote data transmission. Astronomy prioritizes compression performance, followed by manageable encoding times. Slow decoding on the ground is a non-issue. Traditional compression methods optimize for fast decoding, so future work should

explore trading decoding speed for higher compression ratios. This asymmetry favors autoregressive methods that can yield higher compression ratios but slow sequential decoding (Yang et al., 2023b), with Transformer models being potential candidates (Child et al. (2019), Roy et al. (2021)).

**Time-series imaging** has the potential for better compression ratios by exploiting correlations across time. We achieved extremely high compression ratios on the JWST-2D-Res dataset, as the frames were collected back-to-back in time, whereas the SDSS-3DT dataset did not see the same results when compressing multiple frames over time. This may be due to SDSS-3DT images of the same sky location being taken days apart, with atmospheric conditions, moon phase, and other factors introducing too much variance. We encourage more exploration of dataset construction and compression for back-to-back time-series imagery. Such compression will be critical for the operation of the next-generation of wide-field time-domain surveys from space, such as CuRIOS (Gulick et al., 2022).

| Codec | SDSS-2D (800x800) | Hubble (4144x2068) |
|---|---|---|
| IDF | $0.42 \pm 0.01$ | $6.03 \pm 0.24$ |
| L3C | $5.18 \pm 1.04$ | $73.04 \pm 2.36$ |
| PixelCNN++ | $1.48 \pm 1.05$ | $20.49 \pm 0.18$ |
| JPEG-XL max | $3.14 \pm 0.14$ | $87.76 \pm 13.30$ |
| JPEG-XL default | $0.06 \pm 0.002$ | $0.91 \pm 0.07$ |
| JPEG-LS | $0.02 \pm 0.0002$ | $0.316 \pm 0.04$ |
| JPEG-2000 | $0.09 \pm 0.003$ | $1.76 \pm 0.11$ |
| RICE | $0.008 \pm 0.0002$ | $0.12 \pm 0.02$ |
| VDM | $2301 \pm 229.2$ | $33072 \pm 19.8$ |

Table 3: Compression (encoding) runtime (in sec/image) on the SDSS-2D and Hubble datasets. For neural methods, we measure the time for evaluating the likelihood under the model without entropy coding.

**Further exploration of the data specificity vs. generality spectrum** is needed. There is a wide variety of data types in astronomy, such as spectrometry, radio, and imagery data, and within imagery data, ground-based vs. space-based. Moving down this hierarchy: different telescopes, different instruments within telescopes, and different parameters within instruments follow. As an example, an astronomer setting up an observation for the NIRCAM instrument on JWST may select exposure time, readout pattern, wavelength filter, pupil, etc (JWST, 2024). In our study, we purposefully restricted our scope to imagery data of one or a few wavelength filters, but allowed for a wide range of exposure times, readout patterns, and other parameters. We believe this set of data is specialized enough to demonstrate improvements over generic algorithms such as Rice or HCompress, but also accommodates a reasonably wide range of use cases. The right level of specificity will depend on the compression ratios achieved vs. ease of deployment.

We hypothesize that high diversity, multi-modal datasets may be preferable for training compression models, as indicated by strong generalization performance from training on the diverse Keck dataset (see Sec. 4.5). This would make for a multi-purpose compressor that could be a practical successor to JPEG-XL. Eventually, large, multi-modal models and datasets may leverage imaging, spectroscopic and catalogue data all at once.

## 6 CONCLUSION

AstroCompress aims to incentivize the development of astronomy-tuned neural codecs for eventual real-world deployment, by providing datasets that are representative of real use cases. While this work focuses on lossless neural compression, many other machine learning tasks are intimately related. Compression is "bijectively" linked with likelihood estimation by Shannon's source coding theorem (Shannon, 1948). We suggest the use of our dataset in other machine learning for astronomy contexts, such as self-supervised learning for foundation models, semantic search and anomaly detection. Improved lossless and near-lossless neural codecs explored through AstroCompress will likely transfer to other kinds of high resolution, high bit-depth imagery, such as satellite imagery, radio astronomy and biological imaging (as mentioned in Section 1).

### 6.1 LIMITATIONS

Limitations of our dataset include a lack of spectroscopic, radio astronomy or floating-point data. We leave much work to be done on neural methods that can efficiently compress 3D data cubes. Finally, to reduce computation, our compression results for neural compression methods are obtained from evaluating likelihoods under the models without entropy coding. Our relatively inefficient entropy coding implementation shows a negligible overhead compared to bit-rate estimates, and we leave more efficient implementations of the neural compression algorithms to future work.

ACKNOWLEDGEMENTS

This research is based in part on observations made with the NASA/ESA Hubble Space Telescope obtained from the Space Telescope Science Institute, which is operated by the Association of Universities for Research in Astronomy, Inc., under NASA contract NAS 5–26555. HST data are released under the Creative Commons Attribution 4.0 International license. This work is also based in part on observations made with the NASA/ESA/CSA James Webb Space Telescope. The data were obtained from the Mikulski Archive for Space Telescopes at the Space Telescope Science Institute, which is operated by the Association of Universities for Research in Astronomy, Inc., under NASA contract NAS 5-03127 for JWST. This research has made use of the Keck Observatory Archive (KOA), which is operated by the W.M. Keck Observatory and the NASA Exoplanet Science Institute (NExScI), under contract with the National Aeronautics and Space Administration. Funding for SDSS-III was provided by the Alfred P. Sloan Foundation, the Participating Institutions, the National Science Foundation, and the U.S. Department of Energy Office of Science. The SDSS-III website is http://www.sdss3.org/. All SDSS data used herein are considered in the public domain. Data in GBI-16-2D-Legacy, curated and released to the public in FITS format by Maireles-González et al. (2023), makes use of observations the Isaac Newton Group of Telescopes, from Las Cumbres Observatory global telescope network and from The Joan Oró Telescope (TJO).

We would like to thank Justus Will for running additional baseline experiments for the camera-ready version. Stephan Mandt acknowledges support from the National Science Foundation (NSF) under an NSF CAREER Award IIS-2047418 and IIS-2007719, the NSF LEAP Center, by the Department of Energy under grant DE-SC0022331, the IARPA WRIVA program, the Hasso Plattner Research Center at UCI, the Chan Zuckerberg Initiative, and gifts from Qualcomm and Disney.

REPRODUCIBILITY STATEMENT

Our experimental results rely on already published, publicly available compression models. In cases where the architecture was modified to be compatible with new data formats, we were explicit about the alterations described in Sec. 4 and additional architecture and training details in the supplementary material Sec. C. We have also released the code base used for experiments in Sec. C along with the data sets in the abstract.

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

# Supplementary Material

## A    DATASET SUPPLEMENTALS

| Telescope | Dataset ID | Location | Central Wavelength (Å) | Bandpass Filter | Resolution (arcsec / pix) | Image Sizes (pix×px) | Total # Arrays | Total Dataset Size (GB) | Approx. Total Pixels |
|---|---|---|---|---|---|---|---|---|---|
| LCO | GBI-16-2D-Legacy | multiple sites Earth | varies | $B, V, rp, ip$ | 0.58 | $3136 \times 2112$ | 9 | 0.1 | $5.5 \times 10^7$ |
| Keck | GBI-16-2D | HI, USA, Earth | varies, 4500–8400 | $B, V, R, I$ | 0.135 | $2248 \times 2048$; $3768 \times 2520$ | 137 | 1.5 | $7.4 \times 10^8$ |
| Hubble | SBI-16-2D | LEO, Space | 5926 | F606W | 0.05 | $4144 \times 2068$ | 2140 | 69 | $1.8 \times 10^{10}$ |
| JWST | SBI-16-3D | L2, Space | 19840 | F200W | 0.031 | $2048 \times 2048$ $\times$ #Groups | 1273 | 90 | $7.0 \times 10^9$ |
| SDSS | GBI-16-4D | NM, USA, Earth | 3543, 4770, 6231, 7625, 9134 | all of $[u, g, r, i, z]$ | 0.396 | $800 \times 800 \times 5$ $\times$ #Timesteps | 500 | 158 | $1.6 \times 10^{10}$ |

Table 4: Datasets summary. Total pixels computed using each 16-bit data point as one "pixel."

### A.1    NAMING CONVENTIONS

Throughout this work, the data are referred to in three different ways depending on the context; we elucidate the reference conventions here.

#### A.1.1    DATASET ID

The adopted Dataset ID naming convention provides a shorthand description of the origin and form of the data:

$$(origin)(data\ taking\ mode)\text{-}(number\ of\ bits\ per\ pixel)\text{-}(dimensionality),$$

where *origin* is either space-based (SB) or ground-based (GB), and *data taking mode* refers to the primary objective of the exposure, only I (imaging[1]). -2D implies each instance is a 2-dimensional array. -3D may be thought of as a temporal sequence of images (ie., movie) taken in roughly the same portion of the sky. -4D may be thought of as a temporal sequence of images taken in the same portion of the sky at different wavelengths. Clearly, -3D and -4D may be decomposed into individual 2D images. All data in the corpus are `uint16` but future datasets may be added with different bit depths.

This is a standardized way of naming our datasets that include functional details, allowing one to quickly search for or identify the kind of data from the name alone.

#### A.1.2    TELESCOPE OR SURVEY NAME

For brevity and ease of reading, we use the "telescope" column as nicknames to refer to each of our published datasets throughout our paper.

#### A.1.3    EXPERIMENTAL DATASET NAMES

For several reasons, some of our benchmark models were trained on a subset of a dataset. To this end, we added additional descriptors to the dataset nicknames as a way to describe these subselections. LCO, Keck, and Hubble were unchanged and used fully. JWST was used in two forms: a first frame model from JWST-2D (`jwst[t=0]` for every image cube), and JWST-2D-Res (we used all of the residual images to train a separate residual coding model, i.e., `jwst[t=i+1]` − `jwst[t=i]` for

---

[1]Other data taking modes such as spectroscopy may be added in future work.

all $i$, for every image cube). SDSS was split into three datasets: SDSS-2D (`sdss[t=0][`$\lambda$`=2]`),
SDSS-3D$\lambda$ (`sdss[t=0][`$\lambda$`=1,2,3]`), SDSS-3DT (`sdss[t=0,1,2][`$\lambda$`=2]`).

## A.2 ACCESSIBILITY

All datasets herein are released under the AstroCompress project as HuggingFace `datasets` and
may be accessed as `numpy` nd-arrays in Python:

```python
import numpy as np
from datasets import load_dataset

dataset = load_dataset(f"AstroCompress/{name}", config, split=split, \
                       streaming=True, trust_remote_code=True)
ds = dataset.with_format("np", columns=["image"], dtype=np.uint16)
ds[0]["image"].shape # -> (tuple with shape of the numpy array)
```

Here `name` is one of the five datasets described below, `config` is either "tiny" (small number of
files for code testing purposes; default value) or "full" (full dataset), and `split` is one of "train" or
"test". To use the datasets with `pytorch`:

```python
import torch
from torch.utils.data import DataLoader

dataset = load_dataset(f"AstroCompress/{name}", config, split=split, \
                       streaming=True, trust_remote_code=True)
dst = dataset.with_format("torch", columns=["image"], dtype=torch.uint16, \
                          device=device)
dataloader = DataLoader(dst, batch_size=batch_size, num_workers=num_workers)
next(iter(dataloader)) # -> yields the first batch of images
```

where `device` is the pytorch device to use (e.g., "gpu", "mps:0").

## A.3 DATASET COLLECTION AND SCIENTIFIC DETAILS

### A.3.1 DATASET COLLECTION OVERVIEW

All of the data that we have pulled is public data from various astronomical archives. The specific
licenses under which the data have been released are noted in the acknowledgment section. Here we
detail the selection and curation process for each dataset. Table 4 provides a broad overview of the
corpus.

At a high level: for each dataset, we generally come up with a set of logical filters and pull a list
of *observations* via API and/or direct download from the archives where the data are disseminated.
An observation refers to one telescope's "visit" to a certain place in the sky, integrated over a short
period of time (typically less than 20 min). Each observation can result in multiple versions of that
observation in the archives, as many archives store various processed versions of observations along
with the associated calibration data used. For a given observation, we download the one file in raw
form (`uint16`; processed data is usually `float32`). Finally, we select a subset of the downloaded
data that contains no pairwise overlapping of the image footprint on the sky. This ensures that
compression models trained on the training set do not overfit on regions of the sky present in the test
data. It also allows users to safely create their own validation sets from subsets of our training sets.

### A.3.2 GBI-16-2D (KECK)

This is a 2-D optical imaging dataset from the ground-based W. M. Keck Observatory, obtained in
a variety of observing conditions, filters, and across exposure times from 30 seconds to >10 min.
The data are all selected from the Low Resolution Imaging Spectrometer (LRIS; Oke et al. (1995))
and are from scientific observations (as opposed to calibration exposures) obtained by one of the
co-authors of this paper (J.S.B.) from 2005 to 2010. Since LRIS has a dichroic optical element, which
splits the incoming beam at a designated wavelength, the data are obtained either with the blue or

red side camera. Each camera has its own filter wheel allowing for imaging in a variety of different bandpasses. The raw data of a given observation are stored in FITS format (Pence et al., 2010). The LRIS CCDs are readout using 2 amplifiers which are stored in distinct logical and memory sections (called Header Data Units; HDUs) within the FITS files. The blue side camera has two CCDs and the redside camera had 1 CCD before June 2009 and 2 thereafter (Rockosi et al., 2010). In addition to storing the light-exposed regions, LRIS FITS files usually include small regions of virtually read pixels, called overscan regions, which aid image processing.

**Collection**

We used the Keck Observatory Archive (KOA) to identify raw LRIS science images obtained under the principal investigator program by J.S.B. These data were then downloaded and checked for potential footprint overlaps using the positional information of the telescope pointings from the FITS header. The data assembly code for the HuggingFace GBI-16-2D dataset handles the variety of FITS formats and emits 2D images of size $2248 \times 2048$ or $3768 \times 2520$ pix$^2$. An example image from this dataset, with two amplifier reads, is shown in Fig. 1.

### A.3.3 SBI-16-2D (HUBBLE)

This dataset is based on data from the Wide Field Channel (WFC) instrument of the Advanced Camera for Surveys (ACS) onboard the Hubble Space Telescope (HST). Unlike the GBI-16-2D dataset, all observations in SBI-16-2D are obtained in space and with the same bandpass filter (F606W), providing a more uniform point-spread function across the dataset. Our goal with assembling SBI-16-2D is two-fold. First, we aim to provide a large ($> 50$ GB), raw 2-D optical imaging dataset from space. Second, space-based CCD imaging, unlike ground based imaging, suffers significantly from charge particle collisions with the detectors (called "cosmic-ray [CR] hits"). Such random hits add spurious counts to the affected pixels, corrupting the scientific utility of the observations. WFC CCDs also demonstrate large amounts of charge transfer inefficiency leading to correlated streaks in the vertical direction (see Fig. 1).

**Collection**

We first compile a list of observations that fit our criteria, including instrument, filter choice, and integration time ranges. This data was then downloaded from the Mikulski Archive for Space Telescopes (MAST): https://mast.stsci.edu/search/ui/#/hst. For reproducibility, we provide a script that performs this in our HuggingFace repository located at SBI-16-2D/utils/pull_hubble_csv.py. Explanations for each filtering process can be found in the code comments therein. Each FITS file contains two images each of size $4144 \times 2068$; these images are stored in the 1st and 4th headers in the FITS files. After removing overlapping regions in the sky, the final dataset amounts to 4282 images of size $4144 \times 2068$.

### A.3.4 SBI-16-3D (JWST)

This dataset comes from the NIRCAM instrument on the James Webb Space Telescope (JWST). NIRCAM conducts up-the-ramp imaging of various objects in several infrared wavelengths. During up-the-ramp sampling, the aperture is exposed to a region in space, and light continuously accumulates charge on the array. The array repeatedly measures this cumulative charge for multiple frames, thus creating a 3-D time-series image tensor. Every $N$ frames is averaged to create a "group", each of which is stored in our arrays in the $T$ dimension (where $N$ is determined by the "read pattern"). We strongly encourage interested readers to learn more about the JWST readout patterns on the official documentation page [2]. It is worth noting that the time gap between subsequent observations varies, depending on readout pattern, but is roughly in the realm of twenty seconds to two minutes. For this reason, we expect that most physical conditions present in the field of view should remain static across time steps.

**Collection**

Similar to Hubble, we pulled the JWST observations list from the JWST section of MAST. A script version of this can be found at SBI-16-3D/utils/pull_jwst_csv.py. Explanations for

---

[2]https://jwst-docs.stsci.edu/jwst-near-infrared-camera/
nircam-instrumentation/nircam-detector-overview/nircam-detector-readout-patterns

each filtering process can be found in the code comments there. We provide 1273 images of size 2048 by 2048 by $T$, where $T$ represents time, under the F200W wavelength filter.

### A.3.5 GBI-16-4D (SDSS)

This dataset is assembled from "Stripe 82" of the Sloan Digital Sky Survey (SDSS; York et al. (2000)). Stripe 82, a $\sim$250 sq. deg. equatorial region was repeatedly imaged in 5 optical bandpasses with 30 total CCDs over the course of many months for several years, with supernova discovery as the main science objective (Sako et al., 2018). Images are obtained in drift scan mode by fixing the telescope in declination and letting the sky naturally move across the field of view as the CCDs are readout at the sidereal rate. For a given nightly "run" across Stripe 82, the SDSS image reduction pipeline (Stoughton et al., 2002) creates (for each of the 6 camera columns, "camcols") "field" images spanning a similar declination (ie., north-south) and right ascension (RA; ie., east-west). The images served by the legacy SDSS archive[3] are 2048×1489 `uint16` pixels, calibrated with a world coordinate system (WCS) that maps pixel location to sky position.

In total, we release 500 GBI-16-4D cubes as part of this dataset, with an uncompressed size of $158GB$. At a fixed wavelength, the images in time mimic the raw `uint16` time sequence produced by imaging satellites, like Kepler, TESS, and CuRIOS (Gulick et al., 2022). Exploiting the correlations in time (and wavelength) should improve the effective compression compared to the single-slice capabilities.

We note that the gap in time between subsequent timestep images may be many nights, resulting in very different background sky conditions. While the background sky levels across nights are relatively uncorrelated, the pixels containing astrophysical sources exhibit significant correlation.

**Collection**

We queried the SDSS Stripe 82 database to assemble a table of 160k unique Stripe 82 field images deemed to be of top quality (`quality` flag = 3) and then randomly sampled fields and determined the center of the field sky positions. For each such field, we queried the SDSS Stripe 82 database for other images (regardless of `quality`) that overlap the field center and downloaded those images. Since field images from different runs are not aligned in RA, we also downloaded the two fields immediately adjacent and created a stitched-together mosaic of those three images. We then used the WCS in each mosaic image to cut out the same position across all available runs across all available filters.

The assembled data are 4-D cube representations of the same $800{\times}800$ pix$^2$ portion of the sky observed from $t{=}1$ up to $\sim$90 times in $f{=}5$ filters ($u$, $g$, $r$, $i$, $z$). Given the excellent but inherently noisy process of WCS fitting, the $t \times f$ image slices in a given cube are spatially aligned to <1 pix. As a result, for a fixed pixel location in a given cube there are high correlations in the pixel value across time and wavelength. While the effective integration time is identical across all image slices, there are slices that are of higher signal-to-noise than others in the same cube and all have varying background levels. In the few cases where an image slices does not fully overlap the central region of the anchor field, we fill the missing region with values of zero.

### A.3.6 GBI-16-2D-LEGACY

This small ground-based dataset obtained on multiple CCDs across many different telescopes is reproduced from the public corpus released by Maireles-González et al. (2023) and assembled by us in HuggingFace `dataset` format. Our experiments only made use of the subset of data from the Las Cumbres Observatory (**LCO**).

**Collection**

We retrieved all of these files from a download page of Maireles-González et al. (2023) in a `.raw` format and used a script (`GBI-16-2D-Legacy/raw_to_fits.py` in HuggingFace) to convert the images to FITS format. Unlike in the other datasets, we did not check for nor remove potential overlapping images.

---

[3]These images are from the reduction pipeline 40/41 ("rerun") and were part of data release 7 from SDSS. Newer reductions of the same raw data were released with different calibrations in `float32` format.

### A.4 Dataset assembly

**Rejection of overlapping images**   After the initial download of these raw images from various astronomy databases, a significant portion of the necessary data curation is to ensure that the imagery does not overlap on the sky. It is particularly essential to ensure that the spatial footprint of all data in the train set does not overlap with that of the test set. Because we anticipate that future use of this data will split it into validation sets as well, we conservatively ensure that all of our images have pairwise zero overlap.

This was done in two stages, the implementations for which can be found in the `utils/`

**Stage 1.** Before downloading raw imagery, we first download a metadata file that contains a list of observations and the right ascension (RA) and declination (DEC) of each observation. These are akin to standard spherical coordinates $\theta$ and $\phi$. The corresponding pixel for the given RA and DEC varies depending on the data source; it can be the image center pixel, the RA and DEC of the target celestial object of interest, or some other pixel within the image entirely. In order to filter out images in this overlapping set, we ran a hierarchical agglomerative clustering (HAC) algorithm via `sklearn.cluster.AgglomerativeClustering` on a matrix of precomputed pairwise angular distance matrix via `astropy.coordinates.angular_separation`. We made a small modification to the astropy source code in order to allow numpy vectorization. The threshold for clustering was selected as `2*FOV`, where `FOV` was the field of view of any given telescope. Within each cluster, a subset of well-separated images was downloaded.

**Stage 2.** After downloading the data, we were able to use the World Coordinate System (WCS) of each image to map any given pixel in an image to an RA and DEC, using `astropy.wcs`. The Python library `spherical-geometry` is used to compute spherical polygons from the four corners of the image in RA/DEC, and computes overlaps between these polygon objects. In some cases, such as for Hubble, there are two images contained within a single FITS file, so we need to compute spherical polygons for both and then use the union of those two polygons for overlap calculations. Using these algorithms, we further filter out overlapping images within each cluster.

**Code to download raw data from source and filter it** can be found in the `utils` folder of each HuggingFace dataset. We hope these files, in combination with the observation list assembly code will facilitate efforts for future experts to expand on our dataset from the astronomy data sources themselves.

**Disclaimers**: Because the Keck dataset did not have WCS information, we did not run stage 2, and instead used a more stringent clustering threshold in stage 1 and took only one image from each cluster. No filtering was done on the very small Legacy dataset; it also contained no WCS, RA or DEC data.

## B  Baseline Algorithms and Implementation

From any dataset parent folder on HuggingFace, run `python utils/eval_baselines.py 2d` to get 2D evaluations of all algorithms. For JWST and SDSS, this `2d` argument may be changed to several other options that can be found in the command arguments `help` docstring. These options allow for JWST residual ("diffs", as stated in the code) compression, compressing entire 3D tensors for JWST and SDSS, as well as some unique SDSS experiments in which we compressed the top 8 bits and bottom 8 bits separately (`2d_top` and `2d_bottom`). We also attempted with poor results to compress 2D SDSS arrays composed of a single spatial pixel, but all wavelengths and timesteps (`tw`). The exact functionalities are documented in the script.

The script also saves the compression ratio, read time, and write time for every single image into a `.csv` file. We have already performed this and uploaded the CSV files for each dataset to the HuggingFace parent repository. Additionally, running statistics on mean values are printed to the console.

**Implementations** for all the non-neural baseline algorithms were adapted from the Python library `imagecodecs=2024.1.1`. All codecs in this library call the native C codec, specifically: JPEG-XL via `libjxl`, JPEG-LS via `charls`, JPEG-2000 via `OpenJPEG`, and RICE forked from `cfitsio=3.49`.

All the non-neural baseline algorithms natively support 16-bit inputs.

JPEG-XL and JPEG-2000 both use "reversible color transforms" to decorrelate different channels from each other before compressing a multi-channel image. JPEG-LS can only compress each channel as an independent 2D image. RICE is a simple histogram-based codec that flattens N-dimensional arrays before encoding.

In our specific implementations, JPEG-XL and JPEG-2000 supported multi-channel compression directly. To evaluate multichannel arrays using JPEG-LS and RICE, we applied `.reshape((H, -1))` to convert them to single-channel. This was performed only on SDSS data, which resulted in an image height of $H = 800$. For reporting compression ratio, we did not consider the bit-rate cost of transmitting the data shape needed for decoding, as this overhead is negligible for even the smallest image in our dataset.

**Compute hardware**   was used as following: We ran classical compression codecs on a single thread with a `Intel(R) Xeon(R) Silver 4112 CPU @ 2.60GHz` processor and neural compression networks on a single `NVIDIA RTX 6000 ADA`.

## C   DETAILS ON NEURAL COMPRESSION METHODS

In this section, we provide an in-depth look at our experiments and modifications to existing neural compression methods to work with our astronomical image data. Our implementation can be found at https://github.com/tuatruog/AstroCompress.

### C.1   DATA FORMAT

As mentioned in the main text, most neural image compression methods are designed to handle 3-channel 8-bit RGB images, so we made minor modifications to the neural compression methods to handle the 16-bit data of AstroCompress.

At a high level, we experimented with two approaches: (1) adding support for 16-bit input directly; (2) treating the 16-bit input as the concatenation of two 8-bit inputs – the most significant byte (MSB) and least significant byte (LSB). We used the better of the two when reporting results, and generally found approach (2) to perform similarly or better than approach (1). As an example, we list the compression ratios obtained with either approach for **PixelCNN++** in Table 5.

- To implement approach (1), we make the following modifications to support 16-bit input directly: For **IDF**, we change the input scaling coefficient from $2^8$ to $2^{16}$, so that it models the set $\mathbb{Z}/65536$ (instead of $\mathbb{Z}/256$). For **L3C** and **PixelCNN++**, which both use a discretized logistic mixture likelihood model (Salimans et al., 2017), we increase the number of bins from $2^8 - 1$ to $2^{16} - 1$.

- To implement approach (2), we generally convert 16-bit input into 8-bit input while doubling the number of channels. For a 2D (single-frame) image of shape $1 \times H \times W$, this corresponds to treating it as an 8-bit tensor of shape $2 \times H \times W$ where the first channel contains the least significant byte (LSB) and the second channel contains the most significant byte (MSB). For a 3D (multi-frame) image of shape $3 \times H \times W$, this corresponds to treating it as an 8-bit tensor of shape $6 \times H \times W$, where the first three channels contain the LSBs of the original input and the remaining channels contain the MSBs of the original input. The neural compression models are modified accordingly to support the increased number of channels:

  - For **IDF**, we simply doubled the number of input/output channels while keeping the rest of the architecture the same;
  - For **L3C** and **PixelCNN++**, we model 2-channel 8-bit images by using only the "R" and "G" parts of the original RGB autoregressive logistic mixture likelihood model, and we model 6-channel 8-bit images by performing the same autoregressive modeling as the RGB case for the first 3 (LSB) channels, and similarly for the remaining 3 (MSB) channels (so the LSB and MSB channels are modeled separately).

| Experiment | 2-channel 8-bit | 1-channel 16-bit |
|------------|-----------------|------------------|
| LCO        | 1.41            | **2.02**         |
| Keck       | **2.08**        | 1.83             |
| Hubble     | **3.13**        | 2.71             |
| JWST-2D    | **1.44**        | **1.44**         |
| SDSS-2D    | **3.35**        | 1.84             |

Table 5: Comparison of compression ratios for PixelCNN++ using the 2-channel 8-bit format v.s. 1-channel 16-bit format. The better result is highlighted in bold.

## C.2 ARCHITECTURE AND TRAINING DETAILS

**IDF**  We adopted the implementation from Hoogeboom et al. (2019) at `https://github.com/jornpeters/integer_discrete_flows`. The network and training hyper-parameters are also set to be consistent with the default configurations from (Hoogeboom et al., 2019), which we find to yield the best results. We train on patches of $32 \times 32$ with a batch size of 256. We use a learning rate of $1 \times 10^{-3}$ and an exponential decay scheduler with rate 0.999.

**L3C**  We adopted the implementation from Mentzer et al. (2019) at `https://github.com/fab-jul/L3C-PyTorch`. We largely followed the default model configuration provided by (Mentzer et al., 2019) to train the model on 2D image data across all datasets. For 3D experiments, we increased the base convolution filter size and adjusted the latent channel size to 96 and 8, respectively. We trained on $32 \times 32$ patches, with a learning rate of $1 \times 10^{-4}$ and an exponential decay scheduler with rate 0.9.

**PixelCNN++**  We adopted the implementation from `https://github.com/pclucas14/pixel-cnn-pp` and adopted the same model architecture as in the default configuration (5 resnet blocks, 160 filters, 12 logistic mixture components, and a learning rate of $2 \times 10^{-4}$). We also explored different ways of formatting/modeling 8-bit data (converted from 16-bit input), such as training two separate models for the LSB and MSB, or concatenating the LSB and MSB across the width dimension instead of channel dimension, but did not observe significant improvements compared to the basic approach of stacking the LSB and MSB along the channel dimension (as described in Section C.1).

**VDM**  We adopt our implementation from `https://github.com/addtt/variational-diffusion-models/tree/main` which is directly based on the official implementation from Kingma et al. (2021) at `https://github.com/google-research/vdm`. We use a scaled-down version of the denoising network from the VDM paper (Kingma et al., 2021), using a U-Net of depth 4, consisting of 4 ResNet blocks in the forward direction and 5 ResNet blocks in the reverse direction, with a single attention layer and two additional ResNet blocks in the middle. We keep the number of channels constant throughout at 128. We train on patches of size $64 \times 64$ (using Adam and a learning rate of $2 \times 10^{-4}$) and evaluate on patches of size $256 \times 256$ to give our model additional context at test time.

## D ADDITIONAL DATA EXPLORATIONS

**What determines how easily images can be compressed?**  We answer this question via inspiration from a study by Pence et al. 2009, who suggested that the compressibility of astronomy imagery is largely determined by the noise level of background pixels, which are not part of "source" objects such as stars and galaxies. The majority of pixels in our images are background pixels.

Shannon's source coding theorem establishes that the entropy of a data source defines the lower bound for the optimal bitrate. Pence et al. 2009 show that if we assume that background pixels are drawn from a Gaussian distribution $\mathcal{N}(\mu, \sigma^2)$, then the corresponding per-pixel entropy in bits is proportional to $\log_2(\sigma)$. We plot this quantity below, against JPEG-LS bits per pixel.

While background pixels are not exactly drawn from a Gaussian, and JPEG-LS is not a perfect codec, the linear relationship on this plot suggests that background noise levels significantly influence an

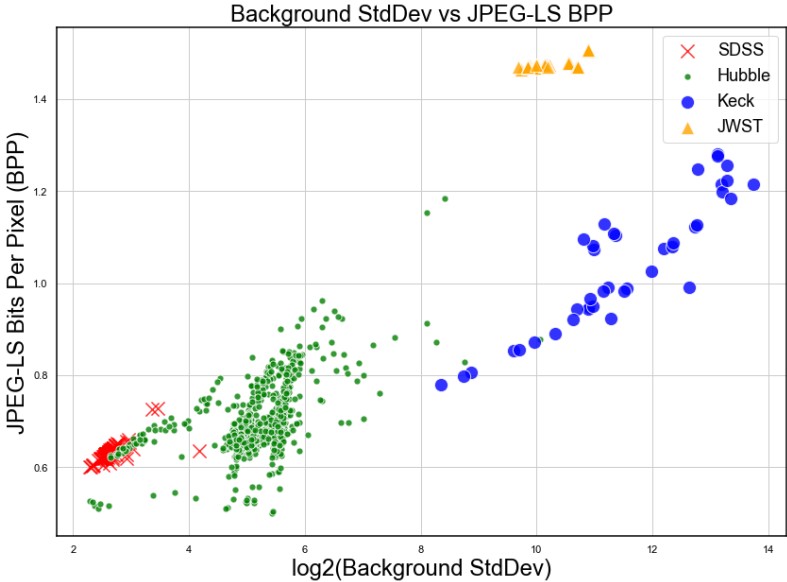

Figure 5: Correlation between background noise variation and compressibility. Each point represents a full-size image. Horizontal axis: log standard deviation of the lower 99 percent of pixel values in a given image. Vertical axis: JPEG-LS bits per pixel, as a representative codec. Single-frame datasets only.

image's compressibility. Moreover, Figure 2 reinforces this insight: as exposure time increases, the number of noise bits in background pixels rises, thereby reducing the image's compression potential.

We note here that the SDSS and Keck datasets show a much tighter linear relationship than Hubble images, suggesting that source pixels are more frequent and more relevant, since we have verified that the background pixels appear quite Gaussian. This corresponds well with intuition, as Hubble is known to sometimes take deep, long exposures of dense stellar fields. JWST images appear as outliers, as it uses HgCdTe detectors for infrared bands, compared to the CMOS detectors for optical imaging of all other telescopes.

## E  EVALUATING PRETRAINED MODELS

We briefly perform a preliminary exploration of how well a neural compression method trained on RGB images generalizes to our astronomy data. Compression models designed for RGB images cannot directly handle our 16-bit data; therefore, we implement a workaround by reformatting each 16-bit pixel as consisting of 3 channels of 8 bits like [MSB, MSB, LSB], where we duplicate the most significant bits (MSB) across the first two channels and fill the last channel with the least significant bits (LSB). As an alternative method, we also tried filling the last channel with only zeros [MSB, LSB, **0**]. We will call these workarounds "duplicated" and "zero-padded," respectively.

We pretrained the L3C model on RGB images from OpenImages (generally around 500 pixels in at least one dimension) and ImageNet64 datasets and evaluated on our single-frame datasets. We also trained IDF on RGB images from Cifar10 dataset and evaluate in the same manner. Overall, we found our RGB-pretrained model to give surprisingly robust performance on astronomy data. When compared to the L3C model pre-trained on the respective astronomy datasets, the RGB-pretrained model did worse by 15% - 25% on KECK, HST, and JWST-2D datasets but actually performed better by 10% - 28% on LCO and SDSS datasets in terms of compression ratio. The detailed results are shown in table 6.

| Model / Dataset | L3C-OpenImages (zero-padded) | L3C-ImageNet64 (zero-padded) | L3C-OpenImages (duplicated) | L3C-ImageNet64 (duplicated) | IDF-Cifar10 (zero-padded) | IDF-Cifar10 (duplicated) |
|---|---|---|---|---|---|---|
| LCO | 1.78 | 2.16 | 2.04 | 2.05 | 1.72 | 1.84 |
| Keck | 1.26 | 1.44 | 1.27 | 1.38 | 0.66 | 0.59 |
| Hubble | 1.77 | 2.18 | 1.88 | 1.90 | 0.59 | 1.08 |
| JWST | 1.08 | 1.19 | 1.06 | 1.18 | 0.62 | 0.58 |
| SDSS | 2.03 | 2.58 | 2.30 | 2.14 | 1.68 | 1.62 |

Table 6: Compression ratio of RGB-trained models across different astronomy datasets.

## F FURTHER MOTIVATION

We present below a brief amount of quantitative evidence on the explosion of the astronomy data scale and the need for advances in its processing:

Astronomy data volumes are growing at an apparently superexponential rate (see Figure 1 of (Maireles-González et al., 2023)). A growth rate of about 10x every 10 years has become nearly 1000x every 10 years. The ability to transmit and store this data will become a massive burden preventing this growth rate from continuing. We hope for advanced compression to alleviate some of this problem.

The current state-of-the-art supercomputers administered by the United States Department of Energy were only recently made ready to handle exascale computing in 2023[4]—specifically Frontier and Aurora, the current two most powerful systems in the world according to `top500.org`. While this "exascale" refers to exaflops rather than exabytes, the need for compute scale is built specifically to address data scale.

In sum, we repeat the breadth of data sources that will soon reach exabyte scale: radio astronomy, satellite imagery[5], genomes, brain mapping and beyond. We note the massive economic potential seen in some of these fields. Lossless methods can be applied without fear, but very carefully crafted lossy designs may soon bring forth orders of magnitude more performant pipelines.

---

[4]`https://science.osti.gov/-/media/bes/besac/pdf/202212/7-Helland--BESAC-Panel.pdf`

[5]`https://www.earthdata.nasa.gov/s3fs-public/2022-02/ESDS_Highlights_2019.pdf`

