# OpenReview forum: "AstroCompress: A benchmark dataset for multi-purpose compression of astronomical data"
_ICLR.cc/2025/Conference — ICLR 2025 Poster_

### Official Review · Reviewer_hYPn · 2024-10-22

**Soundness:** 3
**Presentation:** 3
**Contribution:** 3
**Rating:** 6
**Confidence:** 3

**Summary:**

This paper introduces a benchmark dataset designed for multi-purpose compression of astronomical imagery. A range of image compression codecs is tested and evaluated through comprehensive experiments. The benchmark dataset can serve for researches in astronomy-specific and science-specific lossless neural compression by providing a carefully curated, accessible dataset, along with easily referenceable benchmarks.

**Strengths:**

1. The released dataset is well organized and large in scale and diversity to cover a wide range of astrophysics data.
2. The tested anchors are representative, including both popular learned and conventional codecs.

**Weaknesses:**

1.	Two different methods for handling 16-bit images are discussed and tested. It is recommended that the performance differences between these methods be highlighted.
2.	The proposed dataset’s effectiveness, especially in comparison to existing datasets such as Maireles-González et al. (2023), could be explored more thoroughly.
3.	The performance of RGB-pretrained models on the AstroCompress dataset are suggested to be examined and reported.
4.	The implementation details regarding correlated-frame compression could be clarified further. Specifically, it should be explained whether different frames are compressed independently for each tested anchor, or if a different approach is used.

**Questions:**

As an important aspect in image compression, some preliminary experiments and evaluation on lossy compression are expected.

---

> ### Author Response · Authors · 2024-12-01
> **We appreciate the reviewer’s compliments for our dataset, benchmarks, and feedback (Part 1)**
>
> We appreciate the reviewer’s compliments regarding our diverse dataset and benchmarks, as well as the feedback. We address the concerns point-by-point:
>
> > Two different methods for handling 16-bit images are discussed and tested. It is recommended that the performance differences between these methods be highlighted.
>
> The performance difference between the two formats (2-channel 8-bit v.s. 1-channel 16-bit) seems to be dataset-dependent and mostly an empirical question. For each experiment, we reported whichever approach worked better. We include the compression ratios under both approaches in Table 5, Appendix C.1 of our updated submission.
>
> > The proposed dataset’s effectiveness, especially in comparison to existing datasets such as Maireles-González et al. (2023), could be explored more thoroughly.
>
> We would like to emphasize the significant differences between our dataset and the existing [Maireles-González, 2023] dataset:
> 1. Size: The existing dataset was 1-2 GB, while our additional contribution datasets have a total size of 320 GB.
> 2. Diversity of telescope location: The existing dataset contains a small number of images from five different ground-based telescopes. Our datasets span across ground-based and space-based. Lossless compression is significantly more important for space-based compression, where bandwidth is strictly and severely limited.
> 3. Curation of telescope parameters: In our study, we purposefully restricted our scope to imagery data of a single wavelength filter for many of our datasets, but allowed for a wide range of exposure times, readout patterns, and other parameters. In other words, we have intentionally built a diverse dataset in certain parameters, and a constrained dataset in other parameters. The larger size of our dataset allows us and future scientists to study differences in compression performance as a function of these parameters.
> 4. Train/test split and prevention of data leakage: existing datasets are from unknown locations in the sky that may contain overlapping images. In contrast, we have used sky coordinates to carefully assemble train/test splits that are ready for use by ML practitioners with no overlap between the images in the train and test splits.
>
> > The performance of RGB-pretrained models on the AstroCompress dataset are suggested to be examined and reported.
>
> This is an interesting idea; however, an RGB-pretrained model would not be compatible with the AstroCompress dataset. Specifically, RGB images have three 8-bit channels whereas our dataset entails either a single 16-bit channel or two 8-bit channels, so the models need to be redesigned/retrained for each kind of data and won't work across the two kinds of data. In addition, our data distribution is significantly different from natural images: we have background regions that have very low pixel values, and spherical/ellipsoidal objects that represent stars and galaxies which are many orders of magnitude brighter than the background.
>
> ---
> References
>
> [Maireles-González, 2023] Maireles-González, Òscar, et al. "Efficient Lossless Compression of Integer Astronomical Data." Publications of the Astronomical Society of the Pacific 135.1051 (2023): 094502.

---

> > ### Author Response · Authors · 2024-12-01
> > **We appreciate the reviewer’s compliments for our dataset, benchmarks, and feedback (Part 2)**
> >
> > > The implementation details regarding correlated-frame compression could be clarified further. Specifically, it should be explained whether different frames are compressed independently for each tested anchor, or if a different approach is used.
> >
> > We thank the reviewer for the constructive feedback and clarified our writing accordingly in section 4.2 of our updated submission. To answer the reviewer’s question: we ran three experiments that involved single-frame images, temporally correlated residual images, and data cubes consisting of either temporally or spectrally correlated frames, and our approaches are as follows:
> > 1. For single-frame images (LCO, Keck, Hubble, JWST-2D, SDSS-2D), we compress the frames independently.
> > 2. The 2D JWST-Res dataset involves compressing the differences between subsequent timesteps’ 2D images. The frame differences (residuals) are already created in the construction of the JWST-Res dataset, and these residual frames are compressed independently.
> > 3. SDSS (g, r, i) and SDSS (timestep) are compressions of 3D arrays at once, each of size (500, 500, 3). The 3 channels represent either the 3 wavelength bands or 3 timesteps of the same wavelength band. This 3D tensor is compressed all at once, which works differently for each of the methods presented. For example, PixelCNN++ models the probabilities of each value in this 3D tensor autoregressively (conditioning one channel on the previous ones), while JPEG-2000 and JPEG-XL compress each 3D tensor as if it is a color image (e.g., applying a reversible color transformation to decorrelate the 3 channels and compressing each channel independently).
> >
> > > As an important aspect in image compression, some preliminary experiments and evaluation on lossy compression are expected.
> >
> > We choose lossless compression as a starting point as it has a critical use case in astronomical data compression (it directly affects the amount of raw data that can be sent from telescopes), and the results and impacts are relatively straightforward to evaluate. Lossy compression, on the other hand, introduces additional desiderata beyond the bit-rate, in particular notions of suitable fidelity/distortion metrics in the astronomy context which require extensive discussions and evaluations. For example, in astronomy applications, astronomers are more interested in preserving signal-to-noise ratios, spectra, accurate flux, and other measurements in the image reconstruction, rather than the standard mean-squared error.
> >
> > We would like to emphasize that our work is, in large part, a dataset/benchmark submission. Similar to ImageNet, it is very difficult for progress to happen on ML for scientific data when the data is not easily accessible by ML practitioners. We strongly encourage future work in both lossless and lossy compression, and have included several of our ideas for lossy compression directions in the discussion section of our paper.
> >
> > There exists a strong precedent for this separation between lossless and lossy compression. Here are two pairs of sister papers in astronomy image compression, where one is lossless and one is lossy: [Pence, 2009], [Pence, 2010] and [Maireles-González, 2023], [Maireles-González, 2024]. The same is usually true for most general neural compression papers, due to the many different considerations that arise between the two types of compression.
> >
> > ---
> > References
> >
> > [Maireles-González, 2023] Maireles-González, Òscar, et al. "Efficient Lossless Compression of Integer Astronomical Data." Publications of the Astronomical Society of the Pacific 135.1051 (2023): 094502.
> >
> > [Maireles-González, 2024] Maireles-González, Òscar, et al. "Lossy Compression of Integer Astronomical Images Preserving Photometric Properties." Publications of the Astronomical Society of the Pacific 136.11 (2024): 114506.
> >
> > [Pence, 2009] Pence, W. D., Robert Seaman, and R. L. White. "Lossless astronomical image compression and the effects of noise." Publications of the Astronomical Society of the Pacific 121.878 (2009): 414.
> >
> > [Pence, 2010] Pence, William D., R. L. White, and R. Seaman. "Optimal compression of floating-point astronomical images without significant loss of information." Publications of the Astronomical Society of the Pacific 122.895 (2010): 1065.

---

> > > ### Comment · Reviewer_hYPn · 2024-12-02
> > >
> > > Thank you for the response, some of my concerns have been addressed. But there are some remaining issues:
> > > 1) It is foreseeable that an RGB-pretrained model would be incompatible with the AstroCompress dataset. However, it is still valuable to evaluate how significantly the performance is degraded. Conventional methods, such as JPEG2000, generally perform well across most datasets, but learned methods may not.
> > > 2) When bandwidth is highly constrained, the savings achieved through lossless compression are generally limited. In such scenarios, lossy or near-lossless compression may be preferable. I agree that it is a different aspect compared with lossless compression, but I still believe that as a benchmark paper, more evaluation and results are necessary.

---

> ### Author Response · Authors · 2024-12-04
> **We are glad we were able to address some of your concerns. We will try to answer the ones that remain. (Part 1)**
>
> We are glad we were able to address some of your concerns. We will try to answer the ones that remain:
>
> > It is foreseeable that an RGB-pretrained model would be incompatible with the AstroCompress dataset. However, it is still valuable to evaluate how significantly the performance is degraded. Conventional methods, such as JPEG2000, generally perform well across most datasets, but learned methods may not.
>
> We think the question of generalization is an interesting one that deserves more attention and research in the neural compression community. We followed the reviewer’s suggestion and did preliminary exploration on how well a neural compression method trained on RGB images generalizes to our astronomy data. As we mentioned earlier, compression models designed for RGB images cannot directly handle our 16-bit data; therefore, we implement a workaround by reformatting each 16-bit pixel as consisting of 3 channels of 8 bits like [MSB, MSB, LSB], where we duplicate the most significant bits (MSB) across the first two channels and fill the last channel with the least significant bits (LSB). As an alternative method, we tried [MSB, LSB, 0]. We will call these workarounds “duplicated” and “zero-padded,” respectively.
>
> We pretrained the L3C model on RGB images from OpenImages (generally around 500 pixels in at least one dimension) and ImageNet64 datasets and evaluated on our single-frame datasets. We also trained IDF on RGB images from Cifar10 dataset and evaluate in the same manner. Overall, we found our RGB-pretrained model to give surprisingly robust performance on astronomy data. When compared to the L3C model pretrained on the respective astronomy datasets, the RGB-pretrained model did worse by 15% - 25% on KECK, HST, and JWST-2D datasets, but actually performed better by 10% - 28% on LCO and SDSS datasets, in terms of compression ratio. The detailed results are given in the following table, with a few values missing due to time constraints:
>
> | Dataset / Model   |   L3C-OpenImages (zero-padded) |   L3C-ImageNet64 (zero-padded) |   L3C-OpenImages (duplicated) |   L3C-ImageNet64 (duplicated) |   IDF-Cifar10 (zero-padded) |   IDF-Cifar10 (duplicated) |
> |:------------------|-------------------------------:|-------------------------------:|------------------------------:|------------------------------:|----------------------------:|---------------------------:|
> | LCO               |                           1.78 |                           2.16 |                          2.04 |                          2.05 |                        1.72 |                       1.84 |
> | Keck              |                           1.26 |                           1.44 |                          1.27 |                          1.38 |                        0.66 |                       0.59 |
> | Hubble            |                           1.77 |                           2.18 |                          1.88 |                          1.90 |                      –   |                     –   |
> | JWST              |                           1.08 |                           1.19 |                          1.06 |                          1.18 |                      0.62   |                    0.58   |
> | SDSS              |                           2.03 |                           2.58 |                          2.30 |                          2.14 |                        1.68 |                       1.62 |
>
> Given that the deadline for updating the submission has passed, we will incorporate the complete table of results in the next version of our manuscript.

---

### Official Review · Reviewer_hihc · 2024-10-29

**Soundness:** 4
**Presentation:** 4
**Contribution:** 3
**Rating:** 8
**Confidence:** 4

**Summary:**

The paper describes a new dataset for benchmarking neural compression on astronomical data. The paper lays out how the efficacy of astronomical experiments is highly dependent storage and transmission, and that the fidelity of the data must be preserved, motivating the use of neural lossless compression. The dataset itself is apparently unprecedented in scope for astronomical data with five sub-datasets including a mix of 2D still images as well as spatio-temporal and multispectral hypercubes of high resolution. The paper concludes by testing a series of well know neural and non-neural lossless compression algorithms on the dataset.

**Strengths:**

The paper makes a strong contribution to astrological and neural compression research. The paper lays out clearly how astrological experiments depend on good compression and why a dataset specific to astrological data is necessary. The scope of the dataset is also impressive. It is likely that a researcher working on neural compression, lossless or otherwise, would be interested in findings from testing their method on this dataset. Likewise I can imagine astrological researchers using results on this dataset to decide what methods to deploy. Finally the assembly procedure and details in Appendix A shows that great care was put into the curation of the data.

**Weaknesses:**

The only thing which I would have liked to see more discussion on was the point about runtime/compression ratio. Is there any actionable recommendation here? Also given that JPEG-XL is mostly a reference implementation right now and hasn't been optimized while the neural methods require a GPU, is it  fair comparison to say that it is slower?

**Questions:**

* Any actionable recommendation around neural/classical compression based on results?
* Are resource requirements (GPU, memory) factored into speed results?

---

> ### Author Response · Authors · 2024-12-01
> **We greatly appreciate the reviewer's thorough examination of our paper and appendix**
>
> We are highly grateful for the reviewer’s time spent carefully examining our paper and its appendix. We address any concerns below:
>
> > The only thing which I would have liked to see more discussion on was the point about runtime/compression ratio. Is there any actionable recommendation here? Also given that JPEG-XL is mostly a reference implementation right now and hasn't been optimized while the neural methods require a GPU, is it fair comparison to say that it is slower?
>
> This field of neural scientific data compression is still in its earliest stages. Our actionable recommendation is to prioritize the development of lossless neural methods that consistently outperform traditional codecs in compression ratio, even if they are initially slower and GPU-intensive. Once these methods are mature, the next step would be optimizing them for deployment on specialized hardware, including space-grade systems.
>
> We acknowledge that both our neural methods and JPEG-XL are reference implementations, and neither has been optimized for speed. Therefore, direct runtime comparisons should be interpreted with caution; however, given advances in technology, it is likely that telescopes in the future will be equipped with hardware that supports massively parallel computation and can run neural compression methods. Finally, we point out that compression is also very relevant for ground-based telescopes, which can easily host multi-GPU compute clusters.
>
> We updated our manuscript to reflect these considerations.
>
> > Any actionable recommendation around neural/classical compression based on results?
>
> We have the following recommendations and directions that we are extremely excited to work on for future papers. We outline three key directions inspired by our findings and observations:
>
> 1. Autoregressive models, especially those built on the successful Transformer architecture, have strong potential as the best neural codec for lossless astronomy data compression. These models are inherently faster at encoding due to parallelizability, but slower at decoding, which fits well within the astronomy context where the data needs to be captured and encoded efficiently (e.g., from a space telescope) but has a high compute/time budget for decoding and analysis (e.g., with supercomputers on Earth). Given that many Transformer models are state-of-the-art likelihood estimators [Child et al., 2019; Roy et al., 2021], such a model is likely to match or even beat our newly added VDM performance.
> 2. Our results highlight the potential for near-lossless compression to strike a balance between storage efficiency and data fidelity. This approach involves quantizing data into bins to ensure a strict per-pixel maximum reconstruction error, which we anticipate will be acceptable to astronomers for certain use cases.. We have updated the discussion section of our manuscript to reflect this new focus.
> 3. As shown in Figure 4 of our paper, there are substantial differences in the characteristics of foreground (stars, galaxies, etc.) and background (dark pixels) data. Future methods should exploit these differences by applying distinct compression strategies to each.
>
> > Are resource requirements (GPU, memory) factored into speed results?
>
> Given that running-time efficiency results are very much dependent on hardware and the implementation, we only reported the wall clock time (“speed”) to provide a rough comparison between the different methods run on standard hardware. For reference, our neural methods were run on a single RTX 6000 Ada GPU with 48GB of memory, whereas non-neural methods were run on a single Intel(R) Xeon(R) Silver 4112 CPU @ 2.60GHz. We’ve included these configuration details in section 4.6. Although neither the neural methods (which require GPUs) nor JPEG-XL (which requires high-performance CPUs) can be immediately deployable onboard today’s space telescopes, future efforts on optimizing computational efficiency can soon make them feasible.
>
> ----
> References
>
> [Child et al., 2019] Rewon Child, Scott Gray, Alec Radford, and Ilya Sutskever. Generating long sequences with sparse transformers. arXiv preprint arXiv:1904.10509, 2019.
>
> [Roy et al., 2021] Aurko Roy, Mohammad Saffar, Ashish Vaswani, and David Grangier. Efficient content-based sparse attention with routing transformers. Transactions of the Association for Computational Linguistics, 9:53–68, 2021.

---

> > ### Comment · Reviewer_hihc · 2024-12-02
> >
> > Thanks authors
> >
> > I've reviewed the reviewer comments and author responses and I will maintain my accept rating
> >
> > I think the authors have done a good job curating otherwise disjoint datasets in highly domain-specific formats into something accessible for other researchers to build on. While this has the potential to greatly accelerate research into the compression of astronomical data, it, more importantly, can accelerate all astronomical research by way of helping to develop methods for feasible storage of the requisite data.
> >
> > Furthermore the analysis of existing compression methods on the curated datasets is helpful in understanding what the state-of-the-art is right now. I look forward to seeing how it advances in the coming years.
> >
> > To summarize I think this paper provides a good tool to the research community that may be impactful in multiple areas of science and I think the analysis was sufficient to support the usefulness of the data.

---

> > > ### Author Response · Authors · 2024-12-03
> > > **We are sincerely thankful for your high praise, and very glad that you see the potential value of our work**
> > >
> > > We are sincerely thankful for your high praise, and very glad that you see the potential value of our work. We too hope that this dataset and others can really spark a revolution in practical ML for science. The scale of data in some of these sciences are so massive that they are ripe for ML to attempt disruptions across the board.
> > >
> > > Please let us know if you have any further questions, concerns or comments, we would love to hear any other perspectives of yours!

---

### Official Review · Reviewer_kTqy · 2024-10-30

**Soundness:** 3
**Presentation:** 3
**Contribution:** 2
**Rating:** 5
**Confidence:** 2

**Summary:**

This paper provide a dataset and benchmark for astronomical data compression. The dataset is relatively large (320 GB). Besides, the dataset composes 5 distinct subsets, which covers images of different dimension, resoluton and content types. The authors evaluate the datasets on different lossless compression methods, including IDF, L3C, PixelCNN++, JPEG series and RICE.

**Strengths:**

The dataset composes of diverse images, from 2D to 4D, which is impressive. The selective of neural lossless codec baseline is smart. In fact. IDF, L3C and PixelCNN++ represent three major paradigms of lossless compression: normalizing flow, latent variable model and auto-regressive model. In fact, I can not think of a better choice if I need to choose three most representative lossless image codec.

**Weaknesses:**

One most evident weakness is that the authors evaluate 7 general lossless codec and only 1 astronomical codec (published in 2009). I have two explainations for this evaluation:
* The authors's evaluation is not thorough, many astronomical codecs are omitted.
* The astronomical data compression is not an active research area, and the only reasonable baseline is published 15 years ago.
Either of the explaination makes me hesitate about accepting this paper.

Besides, I am not really sure about how important astronomical data compression is to ICLR community. From the BACKGROUND AND RELATED WORK section in this paper, I find that most papers of astronomical data compression are published in remote sensing and compression related venues, not machine learning related venues (NIPS, ICLR, ICML, CVPR, ICCV, ECCV, etc.). I am not really sure about:
* How many people in ICLR community is interested in astronomical data compression?
* How many people in astronomical data compression community will read ICLR?

**Questions:**

It is the first time that I ever review a benchmark paper. I am not definitely sure about my evaluation. And I am glad to change my mind if other reviewers are positive toward this paper.

---

> ### Author Response · Authors · 2024-12-01
> **We value the reviewer’s praise for our diverse dataset and choice of neural codecs (Part 1)**
>
> We are encouraged by the reviewer’s positive feedback on our dataset size and diversity as well as our careful choice of neural codecs.
>
> We assure the reviewer that our evaluation reflects the SOTA performance in non-neural compression methods for lossless astronomical data compression and that there is recent interest in this field. We address the reviewer’s concerns below, point by point:
>
> > One most evident weakness is that the authors evaluate 7 general lossless codec and only 1 astronomical codec (published in 2009).
> > The authors's evaluation is not thorough, many astronomical codecs are omitted.
>
> We received a similar question from Reviewer iEQb, so we point to our response there, under the question “If there also exists some codecs specially designed for astro data, the authors should also provide some evaluation.”  In summary, we chose to evaluate RICE due to its superior compression performance and widespread adoption in the astronomy field, particularly compared to alternative astronomy codecs like HCOMPRESS. Since the recent astronomy codec by [Maireles-González, 2023] demonstrated performance similar to JPEG-2000, we chose to evaluate JPEG-2000 because it is well-documented and serves as a representative standard non-neural codec from the Maireles-González paper.
>
> > Astronomical data compression is not an active research area, and the only reasonable baseline was published 15 years ago.
>
> While the most representative baseline was published 15 years ago, there has been recent interest in astronomical data compression, for example, the pair of in-depth papers on lossless and lossy astro compression by [Maireles-González, 2023]. We believe there is a strong motivation for renewed interest in this field, given that we are at the cusp of a data explosion “step change” in astronomy. The field is transforming from an era of reasonable data volumes to overwhelming, unmanageable ones.
>
> NASA telescope projects operate on long time horizons. JWST, currently the latest and most advanced space telescope, was planned more than 20 years ago. For this reason, the hardware onboard is very old—only 65 GB of data can be stored. In addition, the CPU onboard is highly limited. Due to the small scale of data and compute onboard space telescopes until now, there has been little capacity for advanced compression codecs.
>
> With that being said, the scale is growing dramatically going forward.
>
> In 2027, NASA will launch the $4.3 billion Nancy Grace Roman Space Telescope. A recent audit of this project published months ago by the Office of the Inspector General (OIG) for NASA [NASA OIG, 2024] emphasizes that the single greatest concern for the Roman project is data transmission issues due to data scale and lack of bandwidth: “Roman is expected to produce an unprecedented amount of data, and there are significant risks with its current plans for transmitting this information back to Earth. The primary network that NASA will use to downlink the data does not have adequate capacity to support Roman’s data requirements.”
>
> Also in 2027, the $3 billion ground-based Square Kilometer Array “is expected to produce 62 exabytes of data each year… this data flood will be transmitted from each telescope at a blisteringly fast 8 terabits per second, signaling the need for extreme data processing.” [Pool, 2020]
>
> These examples underscore the urgent need for new astro compression algorithms. Billion-dollar costs imply that a few percent in improved data compression could mean a gain of hundreds of millions of dollars in value.
>
> ---
> References:
>
> [NASA OIG, 2024] “Audit of the Nancy Grace Roman Space Telescope Project.” NASA Office of the Inspector General, 31 July 2024, oig.nasa.gov/wp-content/uploads/2024/07/ig-24-014.pdf.
>
> [Pool, 2020] Pool, Rebecca. “Drowning in Data.” SPIE, The International Society for Optics and Photonics, 1 May 2020, spie.org/news/photonics-focus/mayjun-2020/square-kilometer-array-big-data.

---

> ### Author Response · Authors · 2024-12-01
> **We value the reviewer’s praise for our diverse dataset and choice of neural codecs (Part 2)**
>
> > Besides, I am not really sure about how important astronomical data compression is to ICLR community. From the BACKGROUND AND RELATED WORK section in this paper, I find that most papers of astronomical data compression are published in remote sensing and compression related venues, not machine learning related venues (NIPS, ICLR, ICML, CVPR, ICCV, ECCV, etc.). I am not really sure about:
>
> > How many people in ICLR community is interested in astronomical data compression?
>
> We acknowledge that astro compression may be “under the radar” for most ML researchers; most existing compression algorithms have been non-neural, and thus have been published in non-AI venues. However, the community is strongly interested in ML applications for science, and we firmly believe our initial foray into ML-based astro data compression demonstrates the promise of this approach and is likely to inspire more interest in the wider ML and astronomy community. We outline the following three-step argument:
>
> 1. At ICLR and other AI venues, there is significant interest in both ML-based compression and more broadly ML for scientific applications. Workshops and tutorials on neural compression have been consistently held at top AI/ML venues in the past few years (e.g., ICLR 2021, NeurIPS 2022, ICML/AAAI/UAI 2023, NeurIPS 2024; see https://neuralcompression.github.io/), and an even larger number of events have been held on the applications of ML/AI to the sciences (see, e.g., https://ai4sciencecommunity.github.io/).
> 2. Scientific data volumes are at an inflection point of exponential growth, from astronomical imaging to biological imaging to genome sequencing, as cited earlier in this rebuttal and in our paper’s introduction. There is a new significant need for advanced codec development, and ML-based methods are currently under-explored. The compression tasks across scientific domains share many similarities:
>     * Need for fast encoding on-device and allowance for slow decoding on-cluster, the opposite of internet compression
>     * Strict requirements for little to no reconstruction error, unlike the more extensive work on neural lossy compression meant for internet media
>     * Highly noisy data
>
> 3. We believe astronomy is an ideal domain for spurring the development of future ML methods because it differs from other scientific data sources like biology in key ways:
>     * Publicly available
>     * Nearly limitless in quantity
>     * Unique financial implications – where percentage improvements in compression can translate to hundreds of millions of dollars in mission value, due to limited storage and the remoteness of space. Backlogged data cannot be “rescued” and thus must be deleted.
>
> > How many people in the astronomical data compression community will read ICLR?
>
> In recent years, there has been a surge of interest in AI for science. Successes in ML being used in weather forecasting and protein folding have served as inspiring examples for other sciences, especially where volumes of data are large. Astronomical data are publicly accessible and vast in quantity. In this context, it is only natural that astronomers are increasingly turning to neural methods to enhance the processing and analysis of astronomical data.

---

### Official Review · Reviewer_iEQb · 2024-11-01

**Soundness:** 3
**Presentation:** 3
**Contribution:** 3
**Rating:** 6
**Confidence:** 4

**Summary:**

This paper introduces AstroCompress, a benchmark dataset designed for multi-purpose compression of astronomical images, providing a foundation for advancing astronomy-specific and science-oriented lossless neural compression. Through comprehensive experiments, various image compression codecs are tested and evaluated.

**Strengths:**

1. The released dataset is made open-accessible and well-organized for researchers to follow.
2. Several lossless compression codecs have been tested including both learned/traditional methods. Evaluations of the impact on datasets and methods are presented.

**Weaknesses:**

1. Since many of the datasets are from publicly accessible resources, therefore i suggest that the authors should contribute more over the analysis and deeper evaluation of the re-organized AstroCompress dataset.
2. The cross-dataset evaluation can be improved. In this paper, only brief conclusions about the generalization of different codecs are provided. A deeper evaluation of the similarity between different datasets and the reason for performing differently for different codecs is suggested.

**Questions:**

If there also exists some codecs specially designed for astro data, the authors should also provide some evaluation.

---

> ### Author Response · Authors · 2024-12-01
> **We appreciate the reviewer’s positive feedback and constructive criticism (Part 1)**
>
> We sincerely appreciate the reviewer’s positive feedback on our dataset’s ease of accessibility, thorough evaluation of compression methods, and constructive criticism. We are happy to address these below, point by point.
>
> > Since many of the datasets are from publicly accessible resources, therefore I suggest that the authors should contribute more over the analysis and deeper evaluation of the re-organized AstroCompress dataset.
>
> While the raw data is publicly available, our work follows in the tradition of landmark datasets like ImageNet by carefully identifying, curating, and transforming complex astronomical images into a cohesive, accessible format for the machine learning community. Data collected by scientific instruments in its most raw form, as is the case for our datasets, is often highly complex and requires domain knowledge to standardize. We outline some of these considerations below:
>
> 1. Astronomical image collection is full of complex parameters and data types, with each survey employing unique imaging patterns over the sky and across time. Writing AstroSQL and the relevant APIs to compile data is nontrivial, partially due to a lack of documentation. Some examples: The SDSS telescope has collected dozens of images of the same patches of sky at different points in time, and it requires extensive processing to find these overlapping images, align them according to star positions, and compile neatly packaged multispectral and temporal data cubes. The Keck survey has changed its data format over time [Keck, 2009], and we have written a script in HuggingFace to standardize it into a single format. JWST contains several kinds of “detector readout patterns,” and we have sub-selected the most relevant ones for science [JWST NIRCam Detector Readout Patterns, 2016]
>
> 2. Telescope optics brim with a variety of detector technologies, wavelength filters, pupils, and other choices that capture different celestial structures. Domain knowledge was critical in selecting these parameters with the correct scope, such that the ML models would be sufficiently general for practical use but not so general that model performance suffers due to edge cases. For example, we carefully selected a specific range of exposure times on the Hubble and JWST datasets to avoid undesirable artifacts of “saturated pixels,” where the CCD detector gets maxed out. An example illustrating optical parameters that had to be selected for our data can be readily seen on this JWST documentation page: [JWST NIRCam Pupil and Filter Wheels, 2016] More details can be found in our “utils/*downloading” files, found in each HuggingFace dataset.
>
> 3. By design, telescopes often capture overlapping images of nearby regions of the sky. It is useful to treat these as different training data points because variation still exists between such images. However, for our 2D single-frame datasets, this presents the risk of an image in the train set overlapping on the sky with an image in the test set.  To avoid this train-test contamination, we developed a robust preprocessing approach to ensure disjoint sky footprints across the train/test splits. This was done by clustering data points by their sky coordinates and assigning each cluster to either train or test, minimizing the amount of data we were forced to discard. This algorithm is outlined in Appendix A.4 and in the HuggingFace “utils/*filtering.ipynb” files of each dataset.
>
> All of our code that processes the raw, messy data from public sources into accessible code is on HuggingFace. To get a better impression of the complexity, we encourage the reviewers to view this code in the utils folders and <dataset_name>.py files of each dataset; we believe the scale of domain knowledge and preprocessing will be readily apparent.
>
> ---
> References:
>
> [JWST NIRCam Detector Readout Patterns, 2016] “NIRCam Detector Readout Patterns.” JWST User Documentation, 2016, jwst-docs.stsci.edu/jwst-near-infrared-camera/nircam-instrumentation/nircam-detector-overview/nircam-detector-readout-patterns#gsc.tab=0.
>
> [JWST NIRCam Pupil and Filter Wheels, 2016] “NIRCam Pupil and Filter Wheels.” JWST User Documentation, Space Telescope Science Institute, 2016, jwst-docs.stsci.edu/jwst-near-infrared-camera/nircam-instrumentation/nircam-pupil-and-filter-wheels#gsc.tab=0.
>
> [Keck, 2009] “LRIS Data Format.” W.M. Keck Observatory, University of Hawaii, 2009, www2.keck.hawaii.edu/koa/public/lris/lris_data_form.html.

---

> ### Author Response · Authors · 2024-12-01
> **We appreciate the reviewer’s positive feedback and constructive criticism (Part 2)**
>
> > The cross-dataset evaluation can be improved. In this paper, only brief conclusions about the generalization of different codecs are provided. A deeper evaluation of the similarity between different datasets and the reason for performing differently for different codecs is suggested.
>
> We organize our response in three points:
>
> **1. Why are some datasets easier to compress than others?**
>
> We completed a new analysis below to help answer this question. As seen in our new Figure 5 in Appendix D, we show that background noise levels, which account for some of the variation between our datasets, are a significant predictor of compressibility. We follow the prior analysis by [Pence, 2009], outlined as follows:
>
> 1. Shannon’s source coding theorem establishes that the entropy of a data source defines the lower bound for the optimal bitrate of lossless compression.
> 2. Most of the pixels in our images are “background” pixels, or not part of an identifiable star or galaxy. To a first approximation, these background pixels are Gaussian noise-dominated.
> 3. [Pence, 2009] show that if we assume that background pixels are drawn from a Gaussian distribution $\mathcal{N}(\mu, \sigma^2)$, then the corresponding per-pixel entropy in bits is proportional to $\log_2(\sigma)$.  We plot this quantity against the bit-rate of JPEG-LS in Figure 5 in our updated manuscript; the figure can also be found [here](https://drive.google.com/file/d/1uaz-2ZU-mr63u2RcU0Q-wd8hKLKRl8RA/view?pli=1).
>
> While background pixels are not exactly drawn from a Gaussian, and JPEG-LS is not a perfect codec, the linear relationship on this plot suggests that background noise levels significantly influence an image's compressibility. Moreover, Figure 2 in our paper reinforces this insight: as exposure time increases, the number of noise bits in background pixels rises, thereby reducing the image's compression potential.
>
> Hubble data points have the weakest linear relationship, and we verified that Hubble’s background pixels do appear Gaussian, suggesting that Hubble images have more “source” pixels than other datasets that cause the apparent variability. This corresponds well with intuition, as Hubble is known to sometimes take deep, long exposures of dense stellar fields. JWST images are outliers in the upper right corner, as JWST uses both different detector technology and wavelength filters: HgCdTe and infrared, compared to the others’ CMOS and visible light.
>
> **2. Why do some train datasets allow our models to generalize better than others?**
>
> We refer the reviewer to Table 2 of our manuscript for experimental results on the generalization performance of IDF, where we train and test on all pairwise combinations of our single-frame datasets. The results show that the cross-data generalization performance heavily depends on the training data used, and the performance difference across different datasets can be quite high and will ultimately depend on the statistical characteristics of the datasets. The Keck train set generalizes exceptionally well.
>
> Referring to the scatterplot above (Figure 5 in our updated manuscript): Keck has the highest background pixel variance of all datasets, the widest range of background pixel variances amongst its images, and a very tight linear correlation between background variance and compressibility. From an astronomer’s standpoint, the Keck images are also the most diverse, as they involve several different wavelength bands.
>
> These facts suggest that a neural model might learn through Keck to focus on the background pixels and generalize out-of-distribution due to the diversity present within the dataset. The JWST-2D and Hubble-2D datasets are ~5x and ~50x  larger than Keck, suggesting that the stronger generalization performance of training on Keck is not a result of a larger training data size.
>
>
> **3. New experiment on cross-data generalizability**
>
> We agree with the reviewer that cross-data evaluation would provide further insight into the behavior of different algorithms on different datasets. To contribute further insight, we conducted a new experiment where we trained IDF on all of the training datasets and evaluated it on each individual dataset. The results in our updated Table 2 show that this model generalizes best, beating the performance of Hubble and SDSS in-domain trained models and nearly matching the others. This suggests potential for a more practical telescope-agnostic neural codec for astronomy data in the future.
>
> ----
> References
>
> [Pence, 2009] Pence, W. D., Robert Seaman, and R. L. White. "Lossless astronomical image compression and the effects of noise." Publications of the Astronomical Society of the Pacific 121.878 (2009): 414.

---

> > ### Author Response · Authors · 2024-12-01
> > **We appreciate the reviewer’s positive feedback and constructive criticism (Part 3)**
> >
> > > If there also exists some codecs specially designed for astro data, the authors should also provide some evaluation.
> >
> > We have indeed evaluated the SOTA codec designed for astronomy image data. Some of us are astro domain experts who use these codecs in our daily work. We also show that many general image codecs and neural codecs outperform the de facto astronomy codec. We give a more detailed explanation below:
> >
> > There exist three astro-specific lossless codecs in the literature: RICE, HCOMPRESS, both developed by NASA before 2009, and one by [Maireles-González, 2023]. However, for the following reasons, we only tested RICE in our experiments:
> >
> > 1. NASA effectively established RICE to be the de facto astronomy compression standard, as it performed similarly to HCOMPRESS and was 3x as fast in their studies [Pence, 2009]. [Maireles-González, 2023] has shown that RICE outperformed HCOMPRESS on five small telescopy datasets.
> > 2. HCOMPRESS was originally designed to be lossy and requires modifications to be lossless.
> > 3. In our experience, RICE is by far the most commonly used codec in the real world.
> > 4. Finally, in [Maireles-González, 2023], the authors’ custom codec performed very similarly to JPEG-2000 (as verified by our evaluations), which is a well-documented, standard codec with robust implementation. So, we chose to evaluate JPEG-2000 and exclude the [Maireles-González, 2023] custom codec.
> > ----
> > References
> >
> > [Pence, 2009] Pence, W. D., Robert Seaman, and R. L. White. "Lossless astronomical image compression and the effects of noise." Publications of the Astronomical Society of the Pacific 121.878 (2009): 414.
> >
> > [Maireles-González, 2023] Maireles-González, Òscar, et al. "Efficient Lossless Compression of Integer Astronomical Data." Publications of the Astronomical Society of the Pacific 135.1051 (2023): 094502.

---

> > > ### Comment · Reviewer_iEQb · 2024-12-02
> > >
> > > Thank you for your response. More explanations have been provided that address some of my concerns. Based on current experiments and evaluation, I tend to keep my rating.

---

> > > > ### Author Response · Authors · 2024-12-03
> > > > **We once again are very grateful for your effort and positive recommendation**
> > > >
> > > > We once again are very grateful for your effort and positive recommendation. Please do not hesitate to let us know if you have any further questions, advice or comments, we would be happy to hear them!
> > > >
> > > > If any of your concerns have not been addressed, please let us know and we will sincerely try again to do so.

---

### Author Response · Authors · 2024-12-01
**General remarks**

We thank all reviewers for their insightful and constructive feedback which has helped strengthen our submission. We are encouraged that reviewers found our dataset “well-organized” [**Reviewer iEQb31**], “unprecedented in scope for astronomical data” [**Reviewer hihc**], “open-accessible and well-organized for researchers to follow” [**Reviewer iEQb**], supported our choice of the “three most representative lossless image codec” which “represent three major paradigms of lossless compression” [**Reviewer kTqy**], and overall valued our paper as “a strong contribution to astronomical and neural compression research” [**Reviewer hihc**].

The reviewers have also raised a few concerns which we address in our revised manuscript. Before we respond to the reviewers’ concerns individually, we summarize our main edits to the manuscript:
1. We performed a new experiment to better understand the cross-data generalization performance of neural compression methods, where we trained IDF on all the single-frame datasets combined. The new results are in Section 4.5 and show that the ensemble approach yields significantly more robust cross-data generalization performance than training on individual datasets alone.
2. We performed a new empirical analysis in Appendix Section D investigating the compressibility of different datasets. Our results reveal that different datasets have widely varying characteristics of noise in the background pixels, and the amount of background noise is intimately related to compression performance across the datasets. This analysis, together with the new experimental results above, contributes a deeper evaluation of the performance of the different codecs on our datasets as requested by **Reviewer iEQb**.
3. In response to **Reviewer hYPn**’s request, we added additional results in Appendix C.1 comparing the compression performance using the two different proposed methods for handling 16-bit data.
4. In response to **Reviewer kTqy**’s concern about the relevance of our contribution, we updated our Introduction section to provide stronger motivation and evidence for the urgent need for research in astronomy data compression. We refer to a recent audit by NASA on the upcoming Roman space telescope project, which raised the issue of data transmission as their single greatest concern [NASA OIG, 2024].
5. To demonstrate the strong potential of neural compression methods, we experimented with a new state-of-the-art likelihood-based model, Variational Diffusion Model (VDM) [Kingma et al., 2021]. While this method is currently too slow to be a practical codec, its estimated compression ratio dominates all the neural and non-neural methods considered on most datasets, often by a wide margin (e.g., 22% improvement on the LCO dataset), which suggests significant room for future improvement. We include our latest results for VDM on the single-frame datasets in the top portion of our updated Table 1 and expect to finish evaluating VDM on the remaining datasets soon.

Lastly, we want to emphasize the broader significance of neural compression for scientific data and our contribution. Scientific data volumes are experiencing exponential growth, creating a critical need for uniquely advanced compression techniques. Unlike internet-focused compression, scientific domains require fast on-device encoding, minimal reconstruction error, and handling of noisy data. Astronomy presents a particularly compelling case due to its publicly available, vast datasets and unique financial implications, where compression improvements can translate to hundreds of millions of dollars in mission value. We believe this work will be of interest to the broader ICLR and AI for science communities, contributing to the ongoing dialogue on machine learning approaches to real-world scientific computing challenges. We hope to spark a research trajectory towards powerful, practical scientific neural codecs deployed in the real world.

---
References

[NASA OIG, 2024] “Audit of the Nancy Grace Roman Space Telescope Project.” NASA Office of the Inspector General, 31 July 2024, oig.nasa.gov/wp-content/uploads/2024/07/ig-24-014.pdf.

[Kingma et al., 2021] Diederik Kingma, Tim Salimans, Ben Poole, and Jonathan Ho. Variational Diffusion Models. NeurIPS, 2021.

---

### Meta-Review · Area_Chair_ZB9q · 2024-12-23

**Metareview:**

This paper introduces a new dataset designed for benchmarking both neural and non-neural compression methods in the context of astronomical data. The dataset is representative of real-world use cases in astronomy and includes a selection of SOTA neural and non-neural methods. Its well-organized structure makes it easy to access and facilitate the benchmarking of new compression algorithms. Reviewers recognized the dataset as a valuable resource for advancing compression techniques in astronomy, benefiting both the machine learning  and astronomy communities. Moreover, the benchmark results highlight the potential of neural compression methods, which could offer research opportunities for ML researchers.

**Additional Comments On Reviewer Discussion:**

One of the major concerns raised by the reviewers was whether the dataset would be of sufficient interest to the ML community. This is a valid concern, which the authors have addressed effectively. Specifically, the dataset includes 2D, 3D, and 4D 16-bit imagery data that has the potential to not only inspire new neural compression algorithms but also open up novel research avenues in astronomy for the ML community. Additional comments focused on the experiments and runtime analysis of the compression algorithms discussed in the paper. Based on the reviewers' responses, I believe the authors have adequately addressed the majority of the reviewers’ concerns during the rebuttal phase.

---

### Decision · Program_Chairs · 2025-01-22

Accept (Poster)